
# Spectral correction of turbulent energy damping on wind LiDAR measurements due to range-gate averaging

Matteo Puccioni and Giacomo Valerio Iungo

Wind Fluids and Experiments (WindFluX) Laboratory, Mechanical Engineering Department, The University of Texas at Dallas, 800 W Campbell Rd, 75080 Richardson, Texas, USA

**Correspondence:** Giacomo Valerio Iungo (valerio.iungo@utdallas.edu)

**Abstract.** Continuous advancements in LiDAR technology have enabled compelling wind turbulence measurements within the atmospheric boundary layer with range gates shorter than 20 m and sampling frequency of the order of 10 Hz. However, estimates of the radial velocity from the back-scattered laser beam are inevitably affected by an averaging process within each range gate, generally modeled as a convolution between the actual velocity projected along the LiDAR line-of-sight and a

weighting function representing the energy distribution of the laser pulse along the range gate. As a result, the spectral energy of the turbulent velocity fluctuations is damped within the inertial sub-range with respective reduction of the velocity variance, and, thus, not allowing to take advantage of the achieved spatio-temporal resolution of the LiDAR technology. In this article, we propose to correct this turbulent energy damping on the LiDAR measurements by reversing the effect of a low-pass filter, which can be estimated directly from the LiDAR measurements. LiDAR data acquired from three different field campaigns

are analyzed to describe the proposed technique, investigate the variability of the filter parameters and, for one dataset, assess the procedure for spectral LiDAR correction against sonic anemometer data. It is found that the order of the low-pass filter used for modeling the energy damping on the LiDAR velocity measurements has negligible effects on the correction of the second-order statistics of the wind velocity. In contrast, its cutoff frequency plays a significant role in the spectral correction encompassing the smoothing effects connected with the LiDAR gate length.

## 1  Introduction

Over the last decades, wind Doppler Light Detection and Ranging (LiDAR) technology has provided compelling features to perform wind turbulence measurements within the atmospheric boundary layer (ABL) for different scientific and industrial pursuits, such as air quality (Trukenmueller et al. , 2004), meteorology (Calhoun et al. , 2006; Emeis et al. , 2007; Grubišić et al. , 2008; Vanderwende et al. , 2015; Horanyi et al. , 2015; Fernando et al. , 2018), aeronautic transportation (George and

Yang , 2012; Schepers et al. , 2012) and wind energy (Iungo et al. , 2013; Iungo and Porté-Agel , 2013, 2014; Iungo , 2016; El-Asha et al. , 2017; Iungo et al. , 2018; Zhan et al. , 2019). In the context of ABL turbulence, scanning Doppler wind LiDARs were assessed against other measurement techniques, such as sonic anemometers and scanning Doppler wind radars, during the eXperimental Planetary boundary layer Instrumentation Assessment (XPIA) campaign (Lundquist et al. , 2017; Debnath et al. , 2017a, b; Choukulkar et al. , 2017; Debnath , 2018).





Different scanning strategies can be designed to characterize different properties of the ABL velocity field through LiDAR measurements (Sathe and Mann , 2013), while the highest spectral resolution is achievable by maximizing the sampling frequency and measuring over a fixed line-of-sight (LOS). Turbulence statistics of the wind velocity field can be retrieved through fixed scans, while providing spectral characterization of the inertial sub-layer (Iungo et al. , 2013) and very large coherent structures present within the ABL (Calaf et al. , 2013). 3D fixed-point measurements can be performed by retrieving the radial

velocity measured simultaneously by three or more LiDARs intersecting at a fixed position (Mann et al. , 2009; Mikkelsen et al. , 2008; Carbajo et al. , 2014, e.g.). Other turbulence statistics can also be evaluated through multiple scanning LiDARs, such as turbulent momentum fluxes (Mann et al. , 2009).

Besides the easier deployment compared to the installation of classical meteorological towers, wind LiDARs currently provide probe volumes, denoted as range gates, smaller than 20 m along the direction of the laser beam and sampling frequency

higher than 1 Hz, which are suitable features for studies on ABL turbulence. A Doppler wind LiDAR allows probing the atmospheric wind field by means of a laser beam, which is back-scattered in the atmosphere due to the presence of particulates suspended in the ABL. The velocity component along the laser-beam direction, denoted as radial or line-of-sight velocity, is evaluated from the Doppler shift on the back-scattered signal. A pulsed Doppler wind LiDAR, like those used for the present work, emits laser pulses to perform quasi-simultaneous wind measurements at multiple distances from the LiDAR as the pulses

travel in the atmosphere. The wind measurements performed over each range gate can be considered as the convolution of the actual wind velocity field projected along the laser-beam direction with a weighting function representing the radial distribution of the energy associated with each laser pulse. Therefore, LiDAR measurements can be considered as the result of a low-pass filtering of the actual velocity field, where the characteristics of the low-pass filter are functions of the energy distribution of the laser pulse over the range gate, length of the range gate and accumulation time (Sjöholm et al. , 2009).

A reduced variance of the wind velocity is generally measured with a Doppler wind LiDAR compared with that measured through a sonic anemometer due to the laser-pulse averaging and different size of the measurement volume. For single-point measurements performed with a Windcube 200S LiDAR and azimuthal angle of the laser beam set equal to the mean wind direction, a variance reduction of $8\%$ was quantified for a gate length of 25 m, while it was increased up to $20\%$ for a probe length of 100 m (Cheynet et al. , 2017).

Attenuation of the measured turbulent kinetic energy due to the averaging over each probe-volume has been recovered by means of a spectral transfer function introduced in Mann et al. (2009). For fixed scans, by leveraging the Taylor's frozen-turbulence hypothesis (Taylor , 1938; Panofsky and Dutton , 1984), the velocity energy spectrum is recovered through the deconvolution of the radial velocity with the weighting function representing the energy of the laser pulse. The critical part of this correction method consists in the empirical definition of the weighting function and its representative length scale (Banakh

and Werner , 2005; Lindelöw , 2008; Mann et al. , 2009). As it will be shown in this paper, corrections performed through this deconvolution procedure often do not provide a satisfactory accuracy for wind turbulence measurements.

In this work, a procedure to correct damping of turbulent kinetic energy associated with wavelengths comparable to the LiDAR range gate is proposed. This procedure is based on inverting the effects of a low-pass filter, whose characteristics are directly determined by the power spectral density (PSD) of the LiDAR measurements. The remainder of this paper is organized



as follows: the theoretical aspects of the correction procedure are discussed in §2, while the experimental campaigns performed to collect the various LiDAR datasets are described in §3. In §4, an assessment of the proposed correction procedure is performed against sonic anemometry, while in §5 the correction procedure is tested for various datasets. In §6, the significance of the filter parameters is discussed by leveraging synthetic turbulent velocity signals. Finally, concluding remarks are reported in §7.

## 2  Correction procedure for the LiDAR velocity spectra

Surface-layer scaling is typically used for spectral models of the wind speed assuming that the velocity integral length-scale is proportional to the height from the ground, $z$, and the Reynolds stresses can be normalized with the square of the friction velocity, $u_\tau$. The Kaimal model is a classical approach to characterize energy spectra of the streamwise wind velocity as a function of the frequency, $f$ (Kaimal et al. , 1972; ESDU , 1985; Kaimal and Finnigan , 1994; Simiu and Scanlan , 1996; NDP

70  , 1998):

$$\frac{fS_u}{u_\tau^2\phi_\epsilon} = \frac{A_f f}{(1+B_f f)^{5/3}}. \tag{1}$$

In this work, $S_u$ is the PSD of the streamwise velocity and $A_f$ and $B_f$ are parameters estimated through best-fitting of the pre-multiplied energy spectra of the LiDAR velocity signals with Eq. 1. The term $\phi_\epsilon$ ($\geq 1$) represents a dimensionless dissipation for non-neutral atmospheric stability regimes, with $\phi_\epsilon$ equal to 1 for neutrally-stratified surface-layer flows (Kaimal et al. ,

1972). For this work, we only consider near-neutral atmospheric conditions and slight variations connected with atmospheric stability are embedded in the coefficient $A_f$ of Eq. 1.

The wind velocity spectra can also be modeled as a function of the non-dimensional frequency $n = fz/U$ ($U$ is the local average horizontal wind speed), as follows:

$$\frac{fS_u}{u_\tau^2\phi_\epsilon} = \frac{A_n n}{(1+B_n n)^{5/3}}, \tag{2}$$

where $A_n$ and $B_n$ are found to be equal to 105 and 33, respectively, for neutrally-stratified flows (Kaimal et al. , 1972). It is noteworthy that for frequencies owning to the inertial sub-layer, the pre-multiplied spectra scale as $f^{-2/3}$, while the maximum value occurs for a frequency equal to $1.5/B_f$ (non-dimensional frequency of $1.5/B_n$).

Considering the Cartesian reference frame, $(x_1,x_2,x_3)$, where the coordinates are aligned with streamwise, transverse and vertical directions, respectively, the wind speed measured by a pulsed Doppler wind LiDAR can be modeled as the convolution

between the projection of the wind velocity $\mathbf{u} = (u_1,u_2,u_3)$ along the laser-beam direction, $\mathbf{n} = (n_1,n_2,n_3)$, with a weighting function, $\phi$, representing the energy distribution of the laser pulse within a range gate (Sjöholm et al. , 2009; Cheynet et al. , 2017; Mann et al. , 2009):

$$v_r(x) = \int_{-l/2}^{l/2} \phi(s)\,\mathbf{u}(s\mathbf{n}+x)\cdot\mathbf{n}ds, \tag{3}$$





where $v_r$ is the radial or line-of-sight velocity measured along the laser-beam direction, $\mathbf{n}$, at a radial distance $x$ from the LiDAR. The length of the range gate is $l$, while $s$ is the radial position within the considered gate. The weighting function, $\phi$, is normalized to unit integral. If the Doppler frequency is determined as the first moment of the signal PSD with the background subtracted appropriately, then the weighting function can be expressed as (Banakh and Werner , 2005; Cheynet et al. , 2017; Mann et al. , 2009):

$$\phi_1(s) = \frac{l - |s|}{l^2} \tag{4}$$

For the LiDAR Windcube 200S, the following weighting function can also be used (Lindelöw , 2008; Mann et al. , 2009):

$$\phi_2(s) = \frac{3(l - |s|)^2}{2l^3} \tag{5}$$

In the spectral domain, the Fourier transform of Eq. (4) is:

$$\varphi_1 = \frac{\sin^2(0.5kl)}{(0.5kl)^2}, \tag{6}$$

while for Eq. 5 is:

$$\varphi_2 = \frac{6}{(kl)^2} \cdot \left[ 1 - \frac{\sin(kl)}{kl} \right], \tag{7}$$

where $k = 2\pi f / U$ is the wavenumber evaluated through the Taylor's frozen-turbulence hypothesis (Taylor , 1938). As shown in Mann et al. (2009), the measured velocity spectrum, $S_L$, can be modeled as:

$$S_L(k_1) = n_i n_j \int_{-\infty}^{+\infty} \int_{-\infty}^{+\infty} |\varphi(\mathbf{k} \cdot \mathbf{n})|^2 \Phi_{ij}(\mathbf{k}) dk_2 dk_3, \tag{8}$$

where: $\mathbf{k} = (k_1, k_2, k_3)$ is the wavenumber vector and summation over repeated indices is assumed. In Eq. 8 , $\Phi_{ij}(\mathbf{k})$ is the spectral tensor obtained as Fourier transform of the Reynolds stress tensor and $\varphi(\mathbf{k} \cdot \mathbf{n})$ is the Fourier transform of the convolution function. When the laser beam stares along the mean wind direction with a relatively low elevation angle, namely with $\mathbf{n} \approx (1,0,0)$, the PSD of the radial velocity, $S_L$, is equal to the product between the spectrum of the actual radial velocity, $\hat{S}_L(k_1)$, and the square of the Fourier transform of the weighting function, $\varphi(k_1)$:

$$S_L(k_1) = \int_{-\infty}^{+\infty} \int_{-\infty}^{+\infty} |\varphi(k_1)|^2 \Phi_{ij}(\mathbf{k}) dk_2 dk_3 = |\varphi(k_1)|^2 \hat{S}_L(k_1). \tag{9}$$

Eq. 9 shows that the spectrum of the measured radial velocity, $S_L$, is equal to the true velocity spectrum, $\hat{S}_L$, low-pass filtered with a certain transfer function. In this work, the latter is modeled as:

$$|\tilde{\varphi}|^2(f) = \left[ 1 + \left( \frac{f}{f_{Th}} \right)^\alpha \right]^{-1}, \tag{10}$$

where $\alpha$ and $f_{Th}$ represent the order and cutoff frequency, respectively, of a low-pass filter (Ogata , 1972). These features of the low-pass filter and, thus, of the LiDAR measuring process, are functions of the LiDAR range gate, elevation angle of the laser





beam, relative angle between wind direction and azimuth angle of the laser beam, accumulation time, and characteristics of the laser pulse. Therefore, it is highly challenging to predict these parameters a priori, while it seems more efficient to estimate $\alpha$ and $f_{Th}$ directly from the specific LiDAR data under analysis. To this aim, we propose the following procedure, which is summarized in the flow chart of Fig. 1. First, the pre-multiplied spectrum of the radial velocity projected in the horizontal mean wind direction is fitted with the Kaimal spectral model of Eq. 1 only for frequencies smaller than $f_{Th}$, which defines a

spectral range with negligible energy damping due to the LiDAR measuring process. As it will be shown in the following, $f_{Th}$ is determined iteratively and is of the order of $\mathcal{O}(10^{-1})$ Hz.

For frequencies higher than $f_{Th}$, the ratio between the fitted Kaimal spectrum and the PSD of the LiDAR velocity, $\varphi_*^2$, is calculated to quantify the effect of the energy damping due to the LiDAR measuring process. Subsequently, $\varphi_*^2$ is fitted with Eq. 10 to estimate the filter order, $\alpha$, and the cutoff frequency, $f_{Th}$. If the guessed cutoff frequency is larger than the fitted

value, the fitting of the LiDAR spectrum with the Kaimal model is repeated by updating the value of the cutoff frequency. Conversely, if the cutoff frequency calibrated on the LiDAR-to-Kaimal ratio is larger than the guessed value, the corrected velocity spectrum, $\tilde{S}_L(f)$, is calculated as:

$$\tilde{S}_L(f) = \frac{S_L(f)}{\tilde{\varphi}^2(f)}. \tag{11}$$

It is noteworthy that, in contrast to existing models using pre-defined functions to correct the energy damping of the velocity

fluctuations, see e.g. Eqs. 6 and 7 (Cheynet et al. , 2017; Sjöholm et al. , 2009), the proposed procedure calculates the charac-

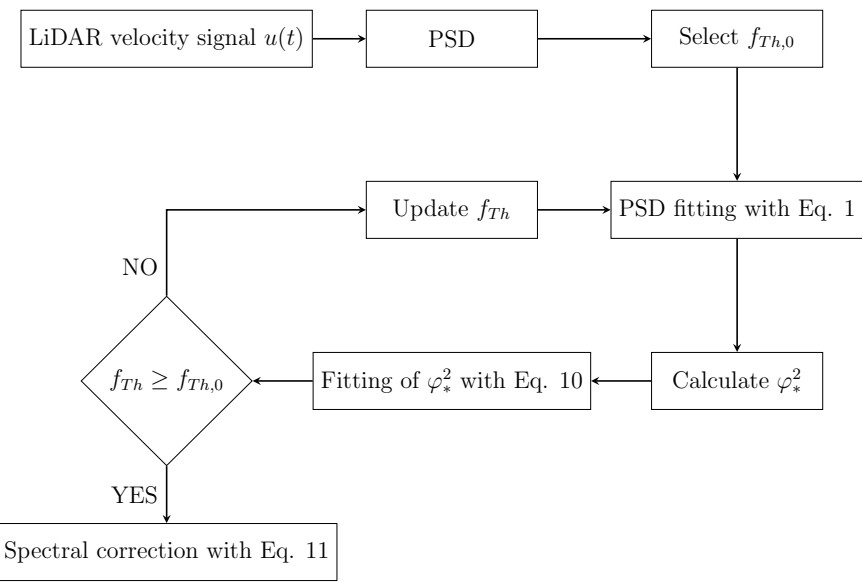

**Figure 1.** Flowchart for the correction procedure of the LiDAR velocity measurements.





teristics of the damping on the LiDAR velocity signals directly from the experimental data, which leads, as it will be shown in the following, to enhanced accuracy in the correction of the LiDAR velocity spectra.

## 3 Experimental Setup and selected LiDAR datasets

The present study is based on wind LiDAR measurements collected from three different experimental campaigns. The first dataset was acquired during the period June 9-24, 2018 at the Surface Layer Turbulence and Environmental Science Test (SLTEST), which is part of the U.S. Dugway Proving Ground facility in Utah (GPS location: $40°08'07.7"N, 113°27'04.6"W$, UTC time offset $-6$ h). Characterized by an elevation variability of 1 m every 13 km (Kunkel and Marusic , 2006; Metzger and Klewicki , 2001), this facility is located in the South West of the Great Salt Lake and extends for 240 km and 48 km along North-South and East-West directions, respectively. An aerial view of the SLTEST facility is reported in Fig. 2(a). During the experiment, the prevailing wind direction was from North-North-East.

The second field campaign was carried out at a test site in Celina, TX (GPS location: $33°17'35.3"$N, $96°49'17.5"$W, UTC time offset $-5$ h), which is a relatively flat terrain with a certain variability in land cover (Fig. 2(b)). For these two field campaigns, wind velocity measurements were performed with a Streamline XR scanning Doppler pulsed wind LiDAR manufactured by Halo Photonics, whose technical details are reported in Table 1. LiDAR staring scans were performed with an elevation angle between $1.98°$ and $10°$, while the azimuth angle was set equal to the mean wind direction. The latter was monitored through Vertical Azimuth Display (VAD) scans with an elevation of $25°$ and sampling period of about 90 s, or through Doppler Beam Swinging (DBS) scans. DBS or VAD scans were executed hourly to monitor variations in the mean wind direction. For the fixed scans, sampling frequency was varied between 0.5 and 3.3 Hz, while the range gate was always set equal to 18 m.

The third campaign considered in this study is the eXperimental Planetary boundary layer Instrumentation Assessment (XPIA), performed during the period March 2- May 31, 2015 at the Boulder Atmospheric Observatory (BAO) research facility

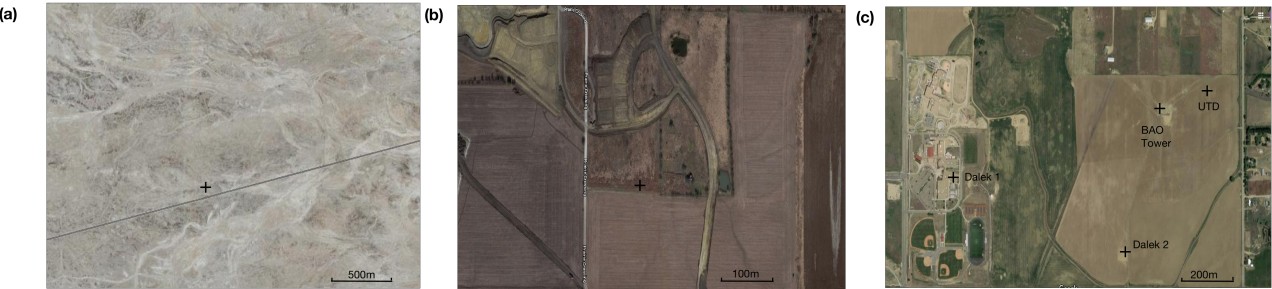

**Figure 2.** Aerial views of the test sites: **(a)** SLTEST facility; **(b)** Celina site; **(c)** site of the XPIA campaign at the Boulder Atmospheric Observatory. Source Google Earth. Black crosses represent the instrument locations.



**Table 1.** Technical specifications of the pulsed scanning Doppler wind LiDARs used for this work, namely a Streamline XR by Halo Photonics and a Windcube 200S by Leosphere.

| Parameter | Value | |
| --- | --- | --- |
| LiDAR | Streamline XR | Windcube 200S |
| Wavelength [$\mu$m ] | 1.5 | 1.54 |
| Repetition rate [kHz] | 10 | 10 |
| Velocity resolution [m s$^{-1}$] | $\pm 0.0764$ | $< 0.5$ |
| Velocity bandwidth [m s$^{-1}$] | $\pm 38$ | $\pm 30$ |
| Number of FFT points | 1024 | - |
| Measurement range | 45 m to 12 km | 50 m to 6 km |
| LiDAR gate length | 18 m to 120 m | 25 m to 100 m |
| Number of gates | 200 | 200 |
| Sampling rate | 0.5 Hz to 3 Hz | 0.1 Hz to 2 Hz |

in Erie, Colorado. For the XPIA campaign, twelve Campbell CSAT3 3D sonic anemometers were mounted on the BAO meteorological tower at heights of 50 m, 100 m, 150 m, 200 m, 250 m and 300 m above the ground. Each height was monitored with two sonic anemometers pointing towards North-West and South-East respectively. Three velocity components and temperature were recorded with a sampling rate of 20 Hz. For a complete description of the scanning strategies and the instruments utilized during the XPIA experiment see Lundquist et al. (2017). Fig. 2c shows the locations of the used LiDARs.

In the present study, from the XPIA experiment we focus on tests performed during the period March 21-24, 2015 with a Windcube 200S scanning Doppler pulsed wind LiDAR manufactured by Leosphere. Technical specifications of the Windcube 200S wind LiDAR are reported in Table 1. The LiDAR performed measurements by staring towards one of the sonic anemometers for a period of 14.5 minutes. A sequential scan at four different heights was done for every hour during each day; details of these tests are available in Debnath (2018). Simultaneous measurements performed with a scanning Doppler wind LiDAR and sonic anemometers are analyzed to assess the proposed spectral correction procedure of LiDAR measurements.

For the Celina and SLTEST field campaigns, the regime of the static atmospheric stability was monitored through sonic anemometers mounted at a height of 3 m in proximity of the LiDAR location. The sampling frequency of the sonic anemometer data was 20 Hz, while atmospheric stability was characterized through the Obukhov length calculated as follows (Monin and Obukhov , 1954):

$$L = -\frac{\theta_v \, u_{\tau,S}^3}{\kappa g \, \overline{w'\theta_v'}}, \tag{12}$$

where $\kappa = 0.41$ is the von Kármán constant, $g$ is the gravitational acceleration, $\overline{w'\theta_v'}$ is the sensible heat flux, $\theta_v$ is the average virtual potential temperature (in Kelvin) and $u_{\tau,S}$ is the friction velocity calculated from sonic anemometer data as (Stull ,





170   1988):

$$u_{\tau,S} = \left( \overline{u'w'}^2 + \overline{v'w'}^2 \right)^{1/4}.$$  (13)

to avoid effects of thermal stratification and buoyancy on our analysis, only datasets acquired under near-neutral conditions are considered, which are selected by imposing the threshold: $|z/L| \leq 0.05$ (Kunkel and Marusic , 2006; Liu et al. , 2017).

The LiDAR velocity signals undergo a quality control process to ensure statistical significance and accuracy of the mea-
surements. For the SLTEST and Celina campaigns, LiDAR fixed scans were performed with the laser beam aligned with the mean wind direction, which is monitored hourly through DBS or VAD scans. Only datasets with variability of the 10-minute averaged wind direction within the range $\pm 20°$ have been considered to avoid significant offset between the LiDAR azimuth angle and the instantaneous wind direction (Hutchins et al. , 2012).

The quality of the LiDAR signals is then checked based on the intensity of the back-scattered signal. For the Windcube 200S
LiDAR, the samples with a carrier-to-noise ratio (CNR) higher than -25 dB are selected, while for the Streamline XR LiDAR data are analyzed only if the intensity of the back-scattered signal is higher than 1.01.

Statistical steadiness of the LiDAR signals is estimated through the non-stationary index (IST) calculated as follows (Liu et al. , 2017):

$$IST = \frac{CV_m - CV_{tot}}{CV_{tot}},$$  (14)

where:

$$CV_m = \frac{1}{N} \sum_{j=1}^{N} CV_j.$$  (15)

In Eq. 15, $CV_j$ is the variance of a signal subset with duration of 5 minutes, while $N$ is the total number of subset signals without overlapping; $CV_{tot}$ is the variance of the signal over the entire period. For Celina and SLTEST campaigns, IST was calculated for 1-hour periods, while for the XPIA deployment the whole 14.5-minutes record was analyzed. For quality control
purpose, signals with $IST \geq 40\%$ are rejected. The IST is calculated for each range gate and the maximum IST value for the selected datasets is reported in the seventh column of Table 2.

Subsequently, a gradient-based procedure is used to remove outliers from the LiDAR radial velocity signals, which are replaced through a least-squares approach in time for each LiDAR gate under investigation. In particular, the estimate of the missing samples is performed with the Matlab function "inpaint". Each signal is linearly detrended in time over the whole duration
to remove large-scale fluctuation, while keeping turbulent velocity fluctuations. Based on the above-mentioned quality-control procedure, five datasets were selected, whose details are reported in Table 2.

The radial velocity $V_r$ measured by a Doppler wind LiDAR, as explained in §2, is expressed as:

$$V_r = V_h \cos(\theta - \theta_w) \cos\Phi + W \sin\Phi$$  (16)

where $\theta$ and $\Phi$ are the LiDAR azimuth and elevation angles, respectively, $\theta_w$ is the wind direction, $V_h$ and $W$ are the horizontal
and vertical wind velocities, respectively. As previously mentioned, for the SLTEST and Celina field campaigns, LiDAR



**Table 2.** Description of the selected datasets: $\Phi$ is the LiDAR elevation angle, $f_S$ is the sampling frequency, IST is the maximum value of non-stationary index, $u_\tau$ is the friction velocity and $z_0$ is the aerodynamic roughness length. The last column reports the symbol used for each dataset.

| Date | Dataset | LiDAR | UTC Time | $\Phi$ [°] | $f_s$[Hz] | max. IST [%] | $u_\tau$ [m s$^{-1}$] | $z_0$ [mm] | Symbol |
|---|---|---|---|---|---|---|---|---|---|
| 23 March 2015 | XPIA | Windcube | 14:15 - 14:29 | 5.00 | 2 | 19.1 | 0.179 | - | - |
| 10 June 2018 | SLTEST | | 10:00 - 13:00 | 3.50 | 1 | 24.6 | 0.414 | $2.1 \cdot 10^{-2}$ | × |
| 26 May 2017 | Celina1 | | 23:35 - 00:35 | 10.00 | 3.3 | 24.1 | 0.479 | 15 | ● |
| 02 October 2017 | Celina2 | Streamline | 22:10 - 01:10 | 5.00 | 0.5 | 37.7 | 0.526 | 87 | ■ |
| 26 January 2018 | Celina3 | | 20:30 - 23:30 | 1.98 | 1 | 39.7 | 0.404 | 17 | ▼ |

measurements were carried out with the azimuth angle equal to the mean wind direction and very low elevation angles (Table 2). Therefore, we can calculate an approximation of the horizontal wind speed as:

$$U_{eq} = V_r / \cos\Phi, \tag{17}$$

which is referred to as horizontal equivalent velocity. In the following $U_{eq}$ is considered to calculate the streamwise velocity
spectrum.

## 4   Assessment of the LiDAR spectral correction against sonic anemometry

In this section, the procedure proposed in §2 to correct the energy damping on the LiDAR velocity measurements due to the energy pulse distribution over a range gate is assessed against sonic anemometry by leveraging the XPIA dataset, whose characteristics are summarized in Table 2. LiDAR fixed scans were performed with an elevation angle of $5°$ to have a range
gate in proximity of a sonic anemometer installed on the BAO tower at height of 100 m. The LiDAR range gate used for that experiment was equal to 50 m, while the sampling rate was set equal to 2 Hz.

    Based on the instantaneous wind direction measured by the sonic anemometer and neglecting the vertical velocity due to the very low elevation angle of the LiDAR laser beam, the horizontal equivalent velocity, $U_{eq}$, is calculated from the LiDAR radial velocity through Eq. 16 and it is reported in Fig. 3 with a blue line by removing its mean value. The PSD of the LiDAR
equivalent velocity, $U_{eq}$, is then calculated and reported in Fig. 4(a) with a grey line.

    In case significant noise in the velocity spectra is observed in proximity of the Nyquist frequency (see e.g. Debnath (2018)), as for this velocity signal, a denoising procedure is then applied to remove possible noise effects on the velocity signals. Following a procedure based on the wavelet transform (To et al. , 2009), the velocity signal is decomposed in a 10-level orthogonal wavelet basis. For each level, a soft-threshold selection is applied to the wavelet coefficients to remove those related





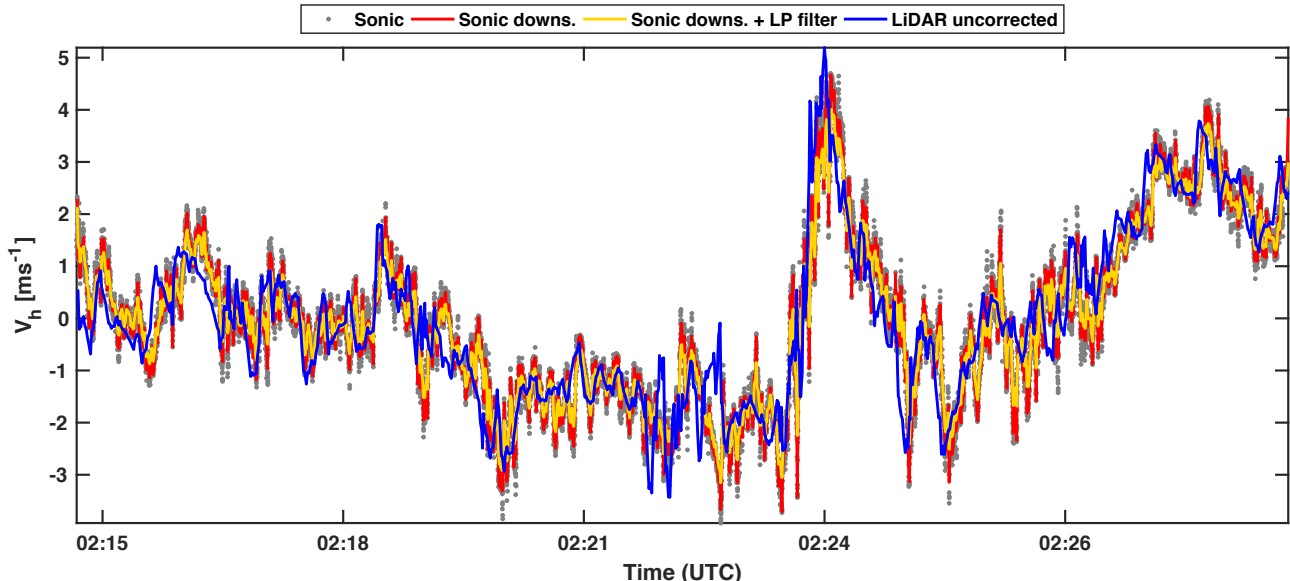

**Figure 3.** Subset of the horizontal velocity measured with a LiDAR and sonic anemometer from the XPIA dataset. Grey dots represent the original 20-Hz-sampled sonic-anemometer data; red line is the sonic anemometer signal down-sampled at 2 Hz; yellow line is the sonic anemometer signal after the convolution of Eq. 10; blue line is the LiDAR signal before the spectral correction.

to noise. The estimated noise-free wavelet coefficients, $d_{jk}$, are calculated as:

$$d_{jk} = \begin{cases} sgn(w_{jk})(|w_{jk}| - T_j) & if \, |w_{jk}| > T_j \\ 0 & otherwise \end{cases}, \tag{18}$$

where $j = 1, ..., 10$ is the number of levels in the wavelet basis; $k = 1, ..., 2^j$ and $w_{jk}$ are the coefficients of the discrete wavelet transform of the original signal. $T_j$ represents a noise-based threshold for the $j^{th}$ level, that for this work is set to (To et al. , 2009):

$$T_j = \frac{med(|w_{jk}|)}{0.67} \cdot \sqrt{2 \log 2^j}, \tag{19}$$

where $med(\cdot)$ stands for the median value. Finally, the denoised signal is reconstructed in time through the modified wavelet coefficients $d_{jk}$. The spectrum of the denoised LiDAR velocity signal is reported in Fig. 4(a) with a light-blue line.

For modeling purpose, the velocity spectra are then smoothed in the frequency domain following the Savitztky-Golay technique (Balasubramaniam , 2005). Specifically, the energy spectrum is smoothed with a moving average over frequency intervals

whose stencil increases with increasing frequency. The result of the averaging process then undergoes a best-fit procedure with a second-order polynomial function. The result is an increased level of smoothness moving towards the Nyquist frequency. The LiDAR velocity spectrum resulting from the de-noising and smoothing procedures is reported in Fig. 4(a) with a blue line. The PSD of the LiDAR velocity is then fitted through the Kaimal model (Eq. 1) for wavenumbers $kl/(2\pi) \leq 0.4$ producing the





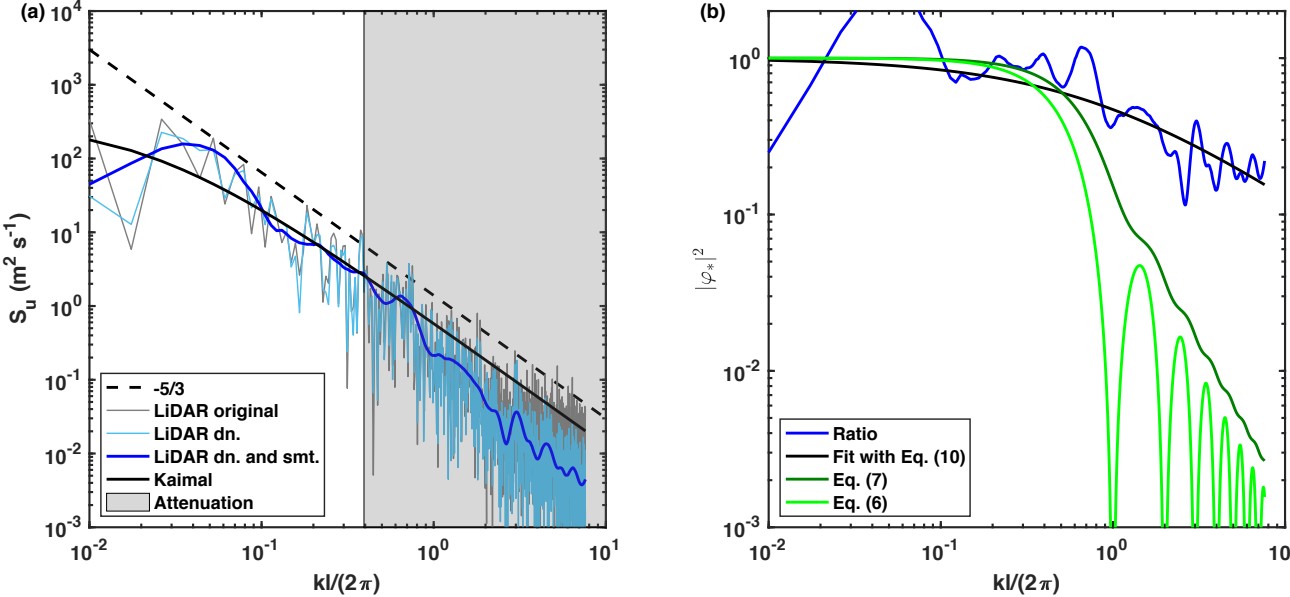

**Figure 4.** Correction of the LiDAR velocity spectrum from the XPIA dataset: **(a)** velocity spectra of the raw LiDAR data (grey), LiDAR data after application of the de-noising procedure (To et al. , 2009) (light blue), LiDAR data after smoothing procedure (dark blue), Kaimal spectrum tuned for $kl/(2\pi) \leq 0.4$ (black) and $-5/3$ slope (black dashed); **(b)** $|\varphi_*|^2$ (blue), $|\varphi_*|^2$ fitted with Eq. 10 (black); predictions from Eq. 6 (dark green) and Eq. 7 (bright green).

following fitting parameters: $A_f = 336.1$ s, $B_f = 355.3$ s, $A_n = 23.3$, $B_n = 24.7$. The resulting Kaimal spectrum is plotted in

Fig. 4(a) with a black line.

A deviation of the LiDAR velocity spectrum from the $-5/3$ scaling of the inertial sub-range is observed for $kl/(2\pi) \gtrsim 0.4$, which is most probably due to the LiDAR measuring process over the range gate. The ratio between the fitted Kaimal spectrum and the LiDAR velocity spectrum, $|\varphi_*|^2$ in Fig. 4(b), is then fitted with Eq. 10 to estimate the low-pass filter of order $\alpha$ and cutoff frequency, $f_{Th}$ . For this LiDAR velocity signal, $\alpha$ is equal to 0.774 and $f_{th}$ equal to 0.15 Hz ( $k_{th}l/(2\pi) \approx 0.8$) with

an $R$-square value of 0.702, which confirms the proposed model being a good approximation for the damping of the velocity fluctuations over the LiDAR range gate. In Fig. 4(b), the weighting functions of Eqs. 6 and 7 are also reported for a range gate $l =$50 m.

To assess accuracy of the estimated low-pass filter in representing the LiDAR averaging process over a range gate, first we apply the estimated low-pass filter to the simultaneous and co-located sonic anemometer velocity signal. The horizontal

velocity retrieved from the sonic anemometer is first down-sampled with the sampling frequency of the LiDAR measurements, namely 2 Hz (red line in Fig. 3, and the respective PSD in Fig. 5(a)). Subsequently, the down-sampled sonic-anemometer signal is low-pass filtered with the filter modeled only using the LiDAR data (yellow line in Fig. 3 and in Fig. 5(a)). The comparison in Fig. 3 of the sonic anemometer signal down-sampled and low-pass filtered with the LiDAR raw signal already





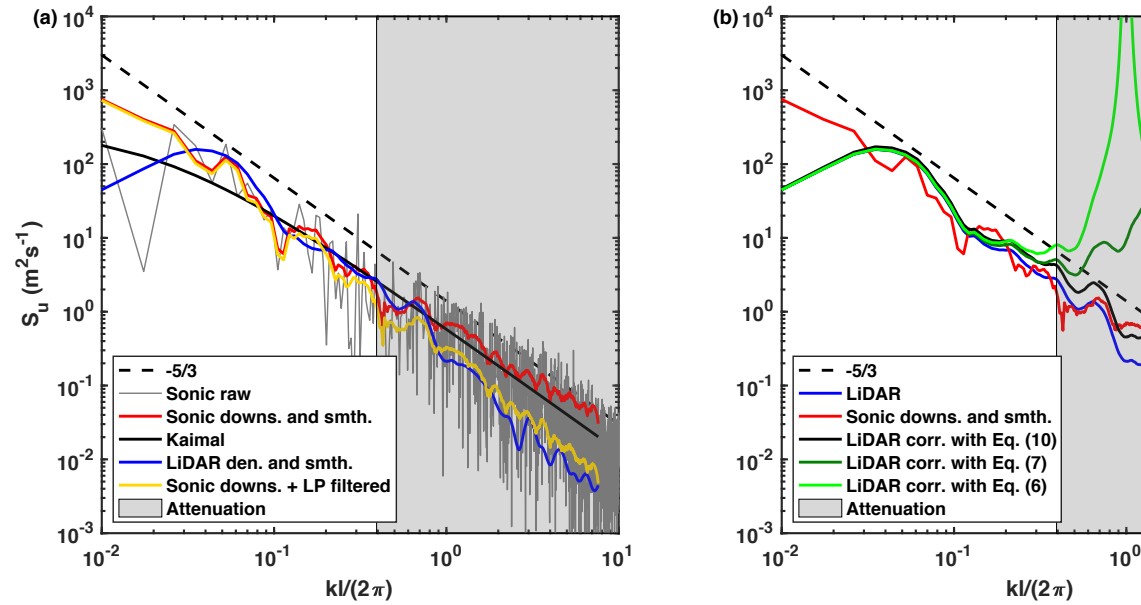

**Figure 5.** Comparison of LiDAR velocity data against sonic anemometry data for the XPIA dataset: **(a)** raw sonic anemometer data (grey), down-sampled and smoothed sonic-anemometer (red), Kaimal model (black) tuned on the LiDAR velocity spectrum (blue), sonic anemometer signal down-sampled and low-pass filtered (yellow); **(b)** LiDAR velocity spectrum (blue), down-sampled and smoothed sonic anemometer spectrum (red), LiDAR spectrum corrected with Eqs. 10, 6 and 7 (black, dark and bright green lines, respectively).

highlights a very good agreement, which suggests that the energy damping carried out by the LiDAR pulse over the range gate
is well represented through the proposed low-pass filter. This feature is further corroborated by the respective spectra reported
in Fig. 5(a). Specifically, the spectrum of the LiDAR signal has the same slope in the inertial sub-range of that for the sonic
anemometer signal down-sampled and low-pass filtered, while some differences are observed for lower frequencies, which are
most probably due to the different size of the measurement volume of the two instruments, namely 50 m for the LiDAR and
0.3 m for the sonic anemometer.

255   The comparison between LiDAR and sonic anemometer data is now presented through a linear regression analysis, which
is reported in Fig. 6. The LiDAR horizontal equivalent velocity, $U_{eq}$, is analyzed against the horizontal wind speed measured
by the sonic anemometer before (Fig. 6(a)) and after (Fig. 6(b)) the low-pass filtering. All the linear regression parameters
improve for the low-pass filtered sonic anemometer data: the slope increases from 0.878 to 0.962, the R-square value increases
from 0.88 to 0.904, and the Pearson correlation coefficient, $\rho$, increases from 0.88 to 0.904.

260   We now aim to correct the LiDAR velocity signal from the energy damping due to the laser pulse distribution over the
range gate. First, the LiDAR velocity spectrum is corrected by using the existing models of Eqs. 6 and 7 with $l = 50$ m, i.e.
the used LiDAR range gate. As shown in Figs. 4(b) and 5(b), these correction methods largely over-estimate the turbulent
energy for frequencies larger than $f_{Th}$. A possible explanation for the poor performance of these deconvolution models could





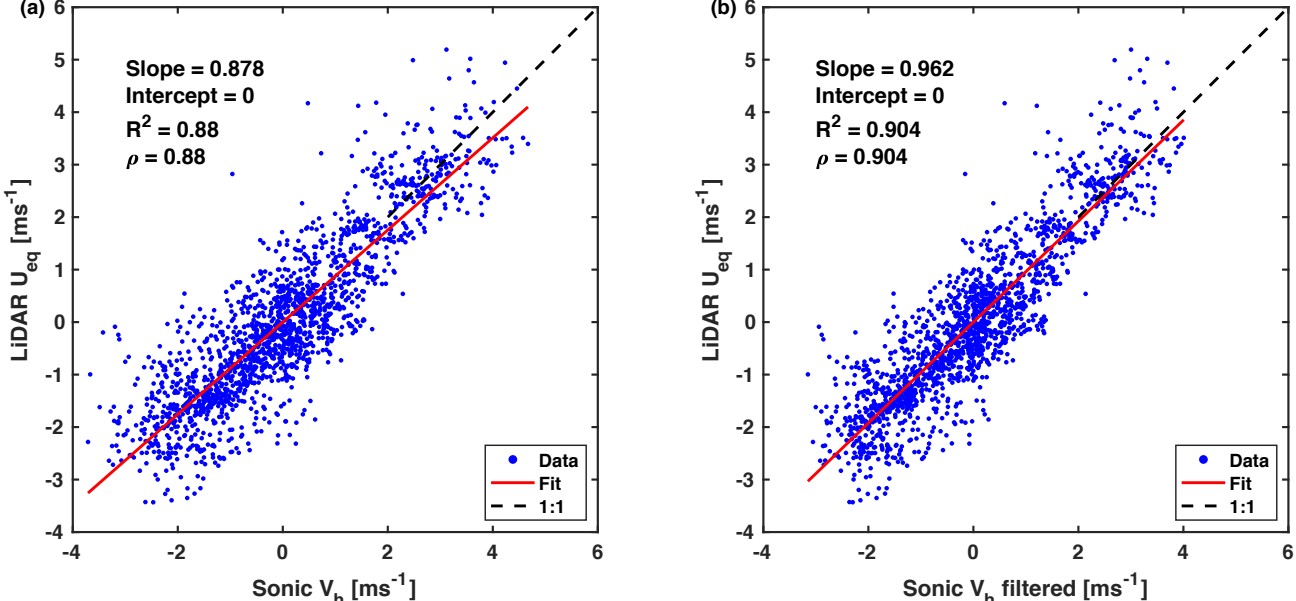

**Figure 6.** Linear regression between LiDAR horizontal equivalent velocity, $U_{eq}$, and sonic anemometer horizontal velocity from the XPIA dataset: **(a)** raw data from the sonic anemometer; **(b)** sonic anemometer data down-sampled and low-pass filtered.

be the different range gate used for the XPIA campaign ($l = 50$ m) in contrast to $l = 30$ m used in the original study of that

deconvolution model (Mann et al. , 2009).

According to the correction technique proposed in this paper, the LiDAR velocity spectrum can now be corrected for the averaging process by reversing the effect of the estimated low-pass filter through Eq. 11. The corrected velocity spectrum is reported in Fig. 5(b) with a black line. The velocity spectrum of the corrected LiDAR velocity signal clearly shows that the expected slope of $-5/3$ in the inertial sub-range is recovered, while the spectral energy for frequencies lower than $f_{Th}$ are

practically unchanged.

To provide more insight on the temporal consistency between the corrected LiDAR time-series and the sonic anemometer data, we band-pass filter LiDAR and sonic anemometer data between the selected cutoff frequency $f_{th} = 0.04$ Hz ($kl/(2\pi) \approx 0.4$) and the Nyquist frequency of the LiDAR measurements, i.e. 1 Hz. Indeed, this is the frequency range more affected by the smoothing process over the range gate and, thus, requiring more correction of the spectral energy. The filtered velocity signals

are subdivided into sub-periods with duration $1/f_{th} = 25$ s, and the standard deviation of the wind velocity is calculated. The linear regression analysis of the velocity standard deviation calculated from the sonic anemometer data and the LiDAR data before and after the spectral correction (reported in Figs. 7(a) and (b), respectively) highlights the positive effect of the spectral correction on the second-order statistics of the LiDAR measurements. For the linear regression of the velocity standard deviation, after the spectral correction the slope increases from 0.674 to 1.083 and the intercept reduces from 0.132 to -0.022,

while the remaining parameters are essentially unaltered.





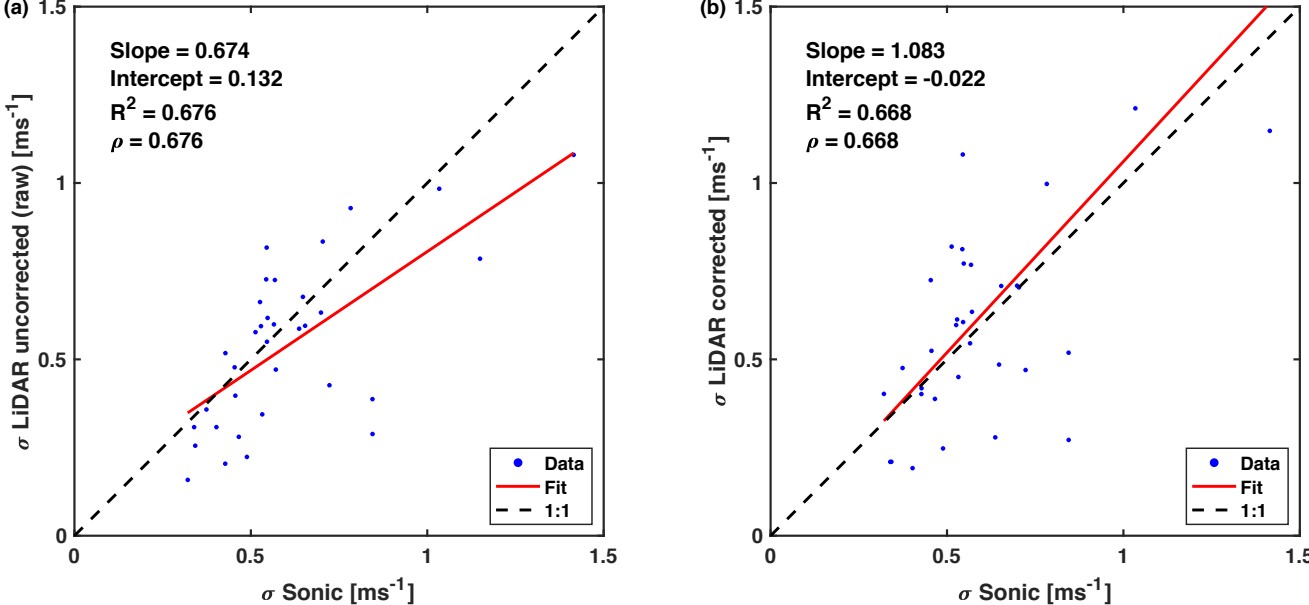

**Figure 7.** Linear regression for the standard deviation of the high-pass filtered velocity signals over 25-s periods: **(a)** sonic anemometer against raw LiDAR data; **(b)** sonic anemometer against corrected LiDAR data.

## 5 Variability of the low-pass filter parameters

For the SLTEST and Celina field campaigns, LiDAR velocity measurements were collected for periods between 2 and 3 hours (see Table 2). The procedure used to obtain the streamwise velocity spectrum is the following: for each LiDAR velocity signal, a high-pass filter is applied with a cutoff frequency of the order $\mathcal{O}(10^{-3})$ Hz to remove low-frequency velocity fluctuations

connected with atmospheric mesoscales. Subsequently, smoothing of the velocity spectra is carried out through the Savitzky-Golay algorithm (Balasubramaniam , 2005).

The first- and second-order statistics of the LiDAR equivalent velocity are plotted in Figs. 8(a) and (b), respectively. For the mean velocity field in Fig. 8(a), while a logarithmic region is generally observed at the lower heights, a noticeable difference in terms of terrain roughness between the SLTEST and Celina sites reflects in different vertical intercepts and respective

aerodynamic roughness length. The latter is estimated to be 0.023 mm for SLTEST and, as average, 37 mm for the Celina site.

The different surface roughness of the two sites also affects the variance of the equivalent horizontal velocity, which is reported in Fig. 8(b) as a function of height. For the datasets collected at the Celina test site, a general increase of the velocity variance is observed with increasing height. Specifically, for the dataset Celina1, after achieving a maximum value at height $z \approx 40$ m, a quasi-logarithmic reduction of the velocity variance is observed with increasing height, which is in agreement

with previous laboratory and numerical studies of canonical boundary layer flows (Kunkel and Marusic , 2006; Meneveau



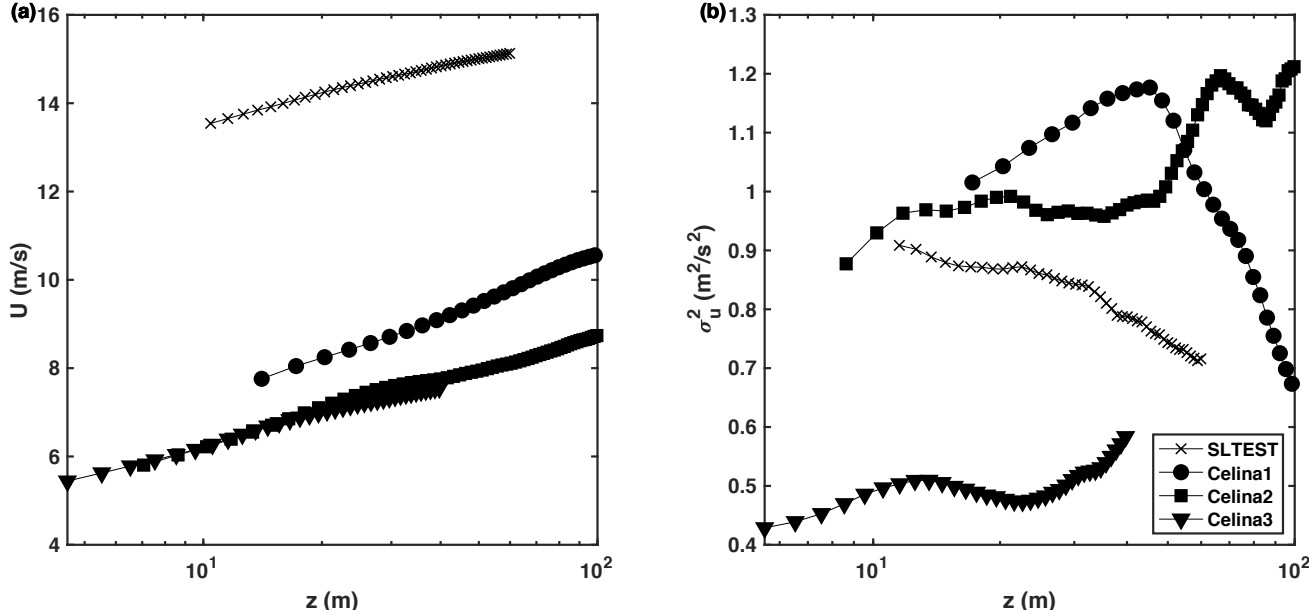

**Figure 8.** First- and second-order statistics of the equivalent velocity, $U_{eq}$, as a function of height for the various datasets: **(a)**: mean value; **(b)**: variance.

and Marusic , 2013). A logarithmic reduction of the velocity variance with increasing height is also observed for the SLTEST dataset throughout the entire height-range.

For the SLTEST dataset, the PSD of the LiDAR velocity signals acquired at the different gates from 10 m up to 60 m with a vertical spacing of 1 m are plotted in Fig. 9. A departure from the expected $-5/3$ slope in the inertial sub-range is observed
starting from $f \gtrsim 0.1$Hz. Considering the observed spectral energy damping, the Kaimal model for the streamwise turbulence (Eq. 1) is fitted on the measured LiDAR velocity spectra only for frequencies with $f \lesssim 0.1$Hz. The fitted Kaimal spectra are reported in Fig. 10(a) for the lowest and highest range gates. The ratio between the LiDAR and Kaimal spectra, $|\varphi_*|^2$, is then calculated and fitted through Eq. 10 to estimate the order and cutoff frequency of the respective low-pass filter (Fig. 10(b)). For the lowest gate at height of 10 m, the fitting procedure estimated $\alpha = 5.08$ and $f_{Th} = 0.14$ Hz, while for the highest gate
at height of 60 m, $\alpha = 4.16$ and $f_{Th} = 0.16$ Hz.

Results of the correction procedure applied to the SLTEST dataset are shown in Fig. 11. It is noteworthy to observe that the fitting of the LiDAR spectra allows detecting the typical decrease in frequency of the energy peak with increasing height (reported with black empty circles in Fig. 11(b)), which is connected with the increase of the integral length-scale (Guala et al. , 2006). The correction of the LiDAR spectra keeps unaltered the energy content of the large energy-containing turbulent
structures, while a significant correction is performed within the inertial sub-range to improve accuracy in the estimate of the turbulent statistics.



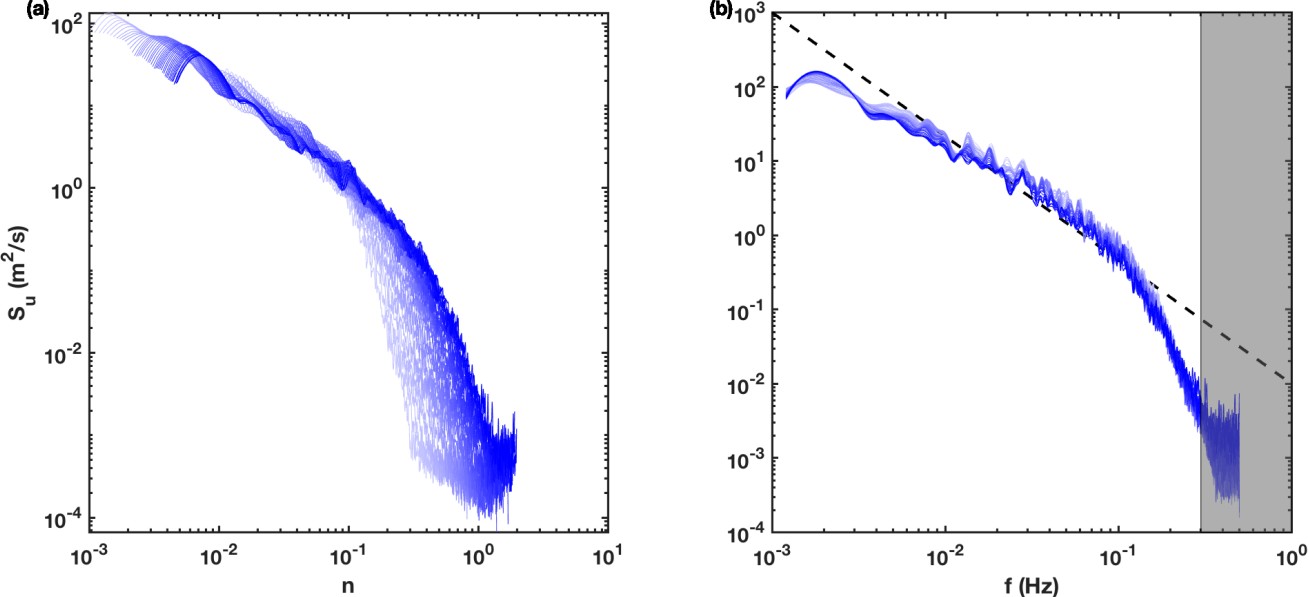

**Figure 9.** Velocity spectra for the SLTEST dataset at different heights: **(a)** spectra as a function of the non-dimensional frequency, $n$: **(b)** spectra as a function of the frequency. Line color is darker with increasing height. Black dashed line represents the $-5/3$ slope. In (b) the shaded area covers the noisy part of the velocity spectra.

Since the correction procedure is based on two consecutive best-fit operations, the robustness of the model is assessed for each LiDAR gate through the R-square value of the respective fitting procedure. Regarding the fitting of the LiDAR spectra with the Kaimal model of Eq. 1, the R-square value is plotted in Fig. 12(a) for all the heights and available datasets. All the

datasets generally show a very good agreement between experimental data and the Kaimal model, i.e. $0.82 \leq R^2 \leq 0.98$, with the SLTEST dataset with the highest level of agreement with $R^2 \geq 0.96$. To quantify accuracy in modeling the actual energy damping due to the LiDAR measuring process through the low-pass filter of Eq. 10, the R-square value of each fitted function is plotted in Fig. 12(b). For this fitting procedure, the R-square values are always larger than $88\%$, corroborating the good approximation for the proposed model.

The proposed spectral correction of the LiDAR measurements is now applied to all the selected datasets. As explained in §2, the first step of the proposed procedure consists in fitting each velocity spectrum with the Kaimal model of Eq. 1. In the right column of Fig. 13 the results of this operation are reported for all the datasets of the SLTEST and Celina field campaigns. The Kaimal model (depicted with red line) has been fitted on the uncorrected LiDAR spectrum (blue lines) using as initial guess $f_{Th,0} = 0.1$ Hz. In the figures, line colors become darker with increasing height. For the sake of clarity, the fitted Kaimal

spectrum is only shown for the highest LiDAR gate.

The second step of the correction procedure consists in approximating the LiDAR-to-Kaimal energy ratio with the low-pass filter of Eq. 10. Firstly, the energy ratio is quantified for each LiDAR range gate, as reported in the left column of Fig. 13. We

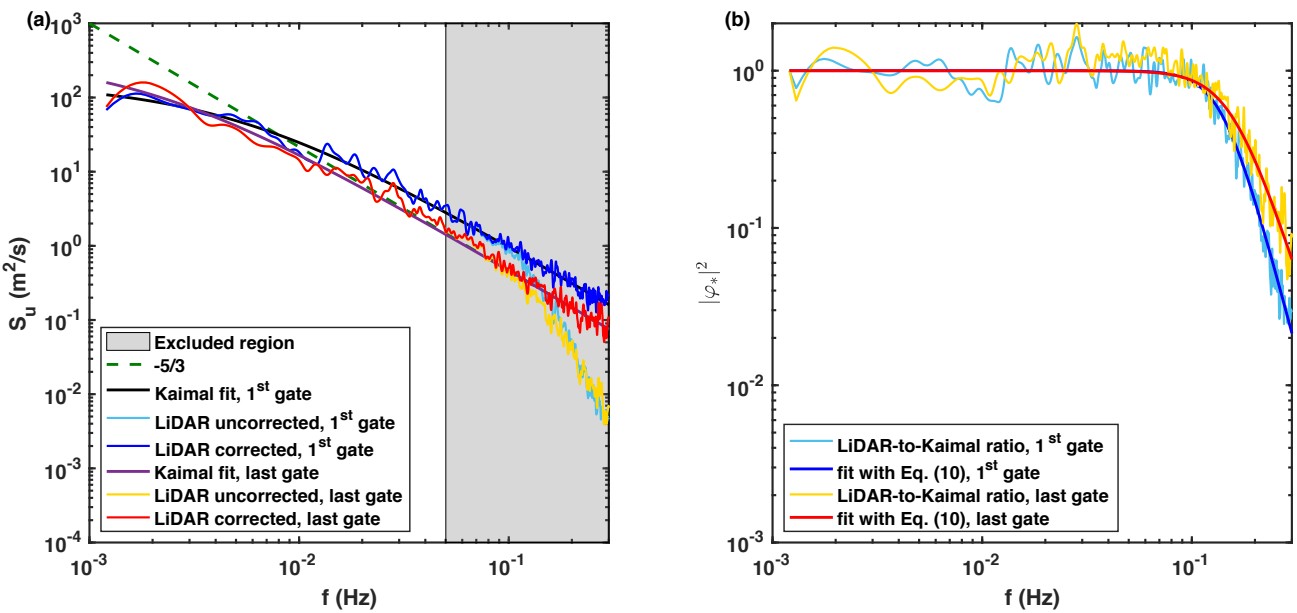

**Figure 10.** LiDAR velocity signals from the SLTEST dataset acquired at $z = 10$ m (first gate, dark and light blue lines) and $z = 60$ m (last gate, red and yellow lines): **(a)** correction of the LiDAR spectra; **(b)** ratio between LiDAR and Kaimal spectra, $\varphi_*^2$, and low-pass filter fitted with Eq. (10).

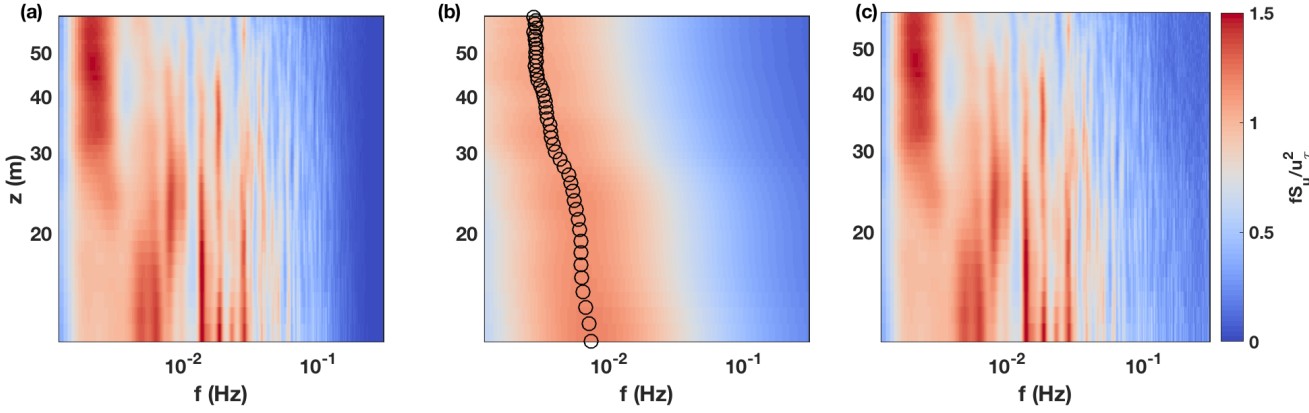

**Figure 11.** Pre-multiplied PSD of the horizontal equivalent wind speed for the SLTEST dataset: **(a)** LiDAR data; **(b)** fitting with the Kaimal model; empty circles represent the peaks of the pre-multiplied spectra. **(c)** corrected spectra.

can observe that the ratio always settles about the unit at the lowest range of the frequency domain, while it monotonically reduces from the cutoff frequency towards the Nyquist frequency, which is clearly an effect of the LiDAR measuring process.



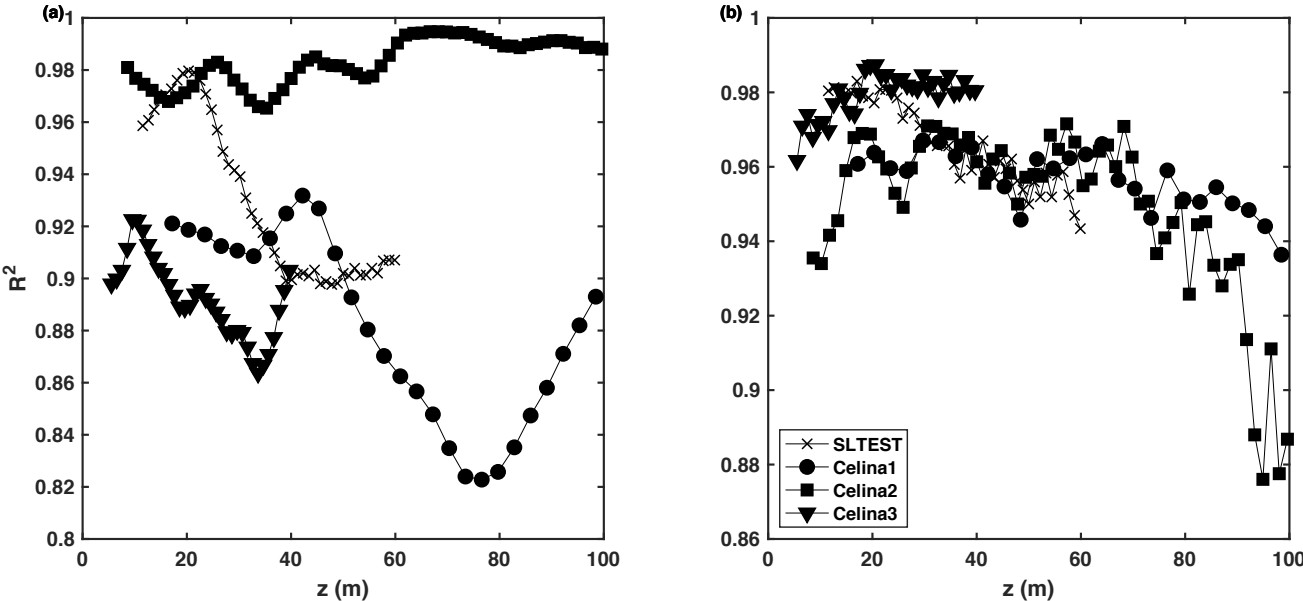

**Figure 12.** R-square parameter of the fitting procedures as a function of height, $z$, for the various datasets: **(a)** Kaimal model (Eq. 1); **(b)** low-pass filter (Eq. 10).

By plotting $|\varphi_*|^2$ as a function of $(f/f_{Th})^\alpha$, a quasi-self-similar behavior is observed among the various range gates, which represents a further assessment of the accuracy of the proposed correction procedure.

As last step, to retrieve the corrected LiDAR velocity spectra, the original spectrum is divided by the modeled correction function, $|\tilde{\varphi}|^2$ (Eq. 11). These corrected LiDAR velocity spectra are reported on the right column of Fig. 13, where we can observe that the $-5/3$ slope of the inertial sub-range is always recovered. The LiDAR spectra corrected with the models of

Eq. 6 and 7 are also reported to highlight the improved accuracy achieved through the proposed method. In particular, it is observed that the existing models of Eqs. 6 and 7 always under-estimate the spectral energy attenuation.

The corrected variance of the LiDAR velocity signals is compared with the respective quantity calculated for the raw LiDAR data in Fig. 14(a); as expected, the wall-normal profile of variance as integral of the de-convoluted spectrum results considerably larger than the same statistic obtained from the convoluted spectrum, meaning that the underestimation related to the probe

resolution is found to be significant. To quantify the effects of the spectral correction on the LiDAR data, the relative percentage increment of variance is calculated from the smallest frequency up to the noise-free high-frequency content, as in Cheynet et al. (2017):

$$\epsilon_\% = \frac{\sigma_C^2 - \sigma_U^2}{\sigma_C^2} \cdot 100, \tag{20}$$

where $\sigma_C^2$ and $\sigma_U^2$ are the corrected and uncorrected, respectively, streamwise velocity variance. The parameter $\epsilon_\%$ is reported

as a function of height in Fig. 14(b). The underestimation in the velocity variance through the LiDAR measurements seems to



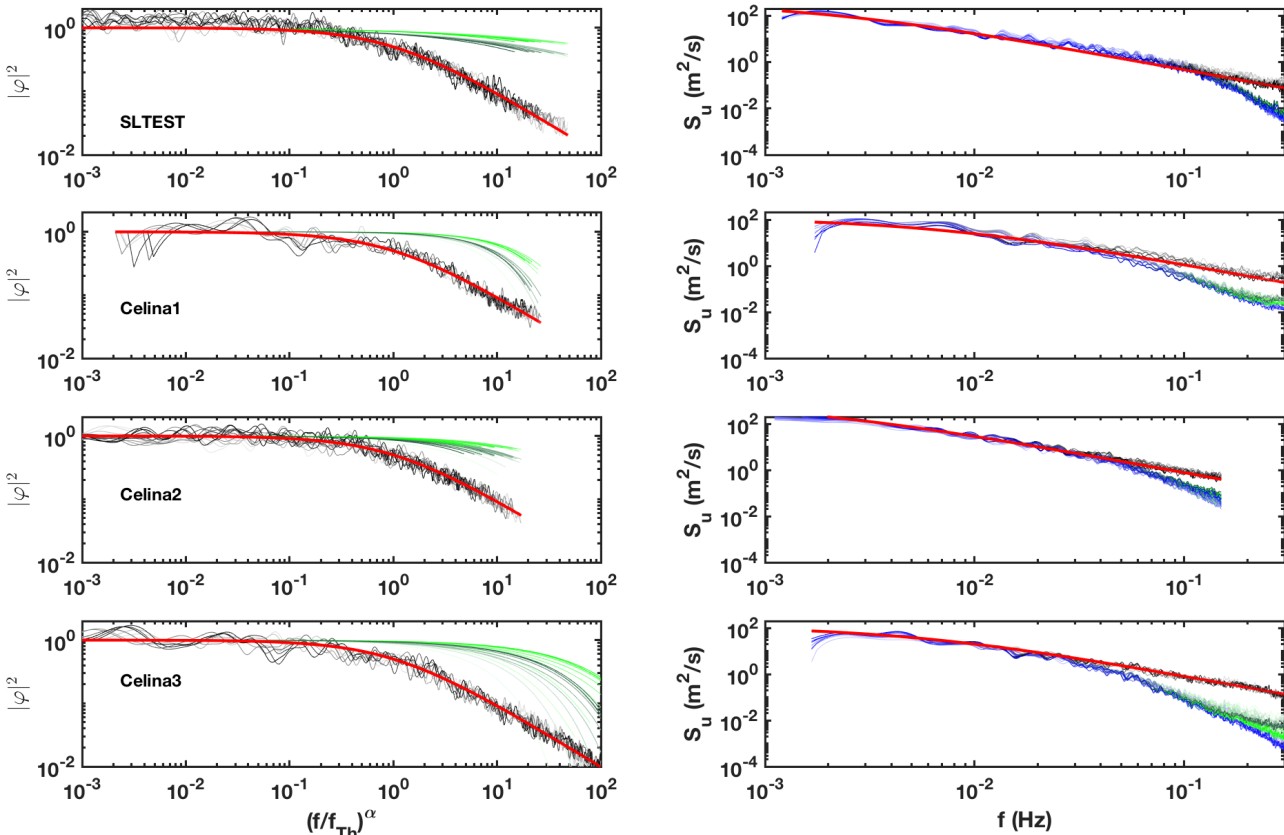

**Figure 13.** Correction of the LiDAR spectra. Left column: black lines are $\varphi_*^2$, red line is the fitted low-pass filters of Eq. (10) for the highest gate. Dark and light green lines represent the convolution function predicted by Eqs. (6) and (7). Right column: blue lines are raw LiDAR spectra, dark and light green lines are LiDAR spectra corrected with Eqs. (6) and (7), respectively, red line is the fitted Kaimal spectrum for the highest range gate and black lines are the LiDAR spectra corrected with the proposed procedure of Eq. 11. Line colors become darker with increasing height.

change with the wall-normal location for SLTEST and the highest portion of Celina1 ($z > 30$m); for the remaining datasets, the percentage error does not change remarkably with height. To clarify this aspect, in the next section we will leverage synthetic turbulent velocity signals to better understand the variability of the LiDAR averaging effect with height.

We now focus on the variability of the parameters of the low-pass filter in Eq. 10 among the various datasets. First, the order
of the low-pass filter, $\alpha$, which is reported in Fig. 15(a), is found to be roughly constant over the height for the datasets Celina1 and Celina3, while it decreases with height for the SLTEST and Celina2 datasets. The cutoff frequency of the low-pass filter, $f_{Th}$, is practically constant with height, as shown in Fig. 15(b).In Fig. 15(c), the equivalent cutoff wavelength, $\lambda_{Th}$ (obtained via Taylor frozen hypothesis) is reported as ratio of the gate length.

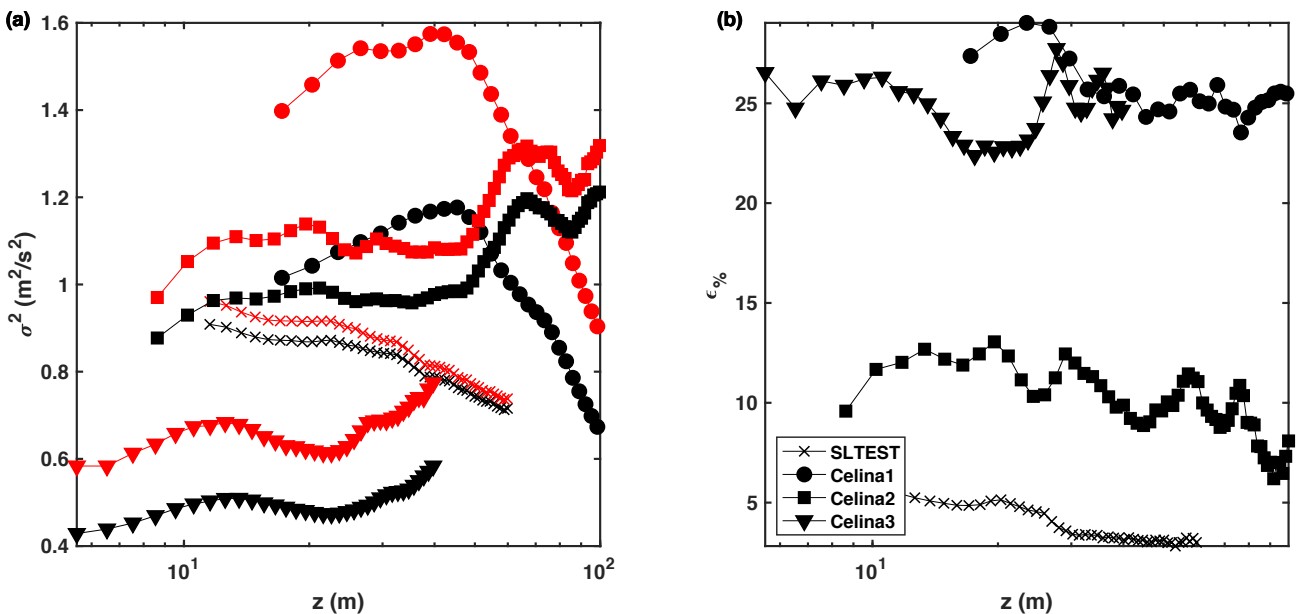

**Figure 14.** Correction of the second-order statistics: **(a)** Variance before (black) and after (red) the LiDAR spectral correction; **(b)** percentage increment of the velocity variance ($\epsilon_\%$) calculated on the retrieved frequency bandwidth.

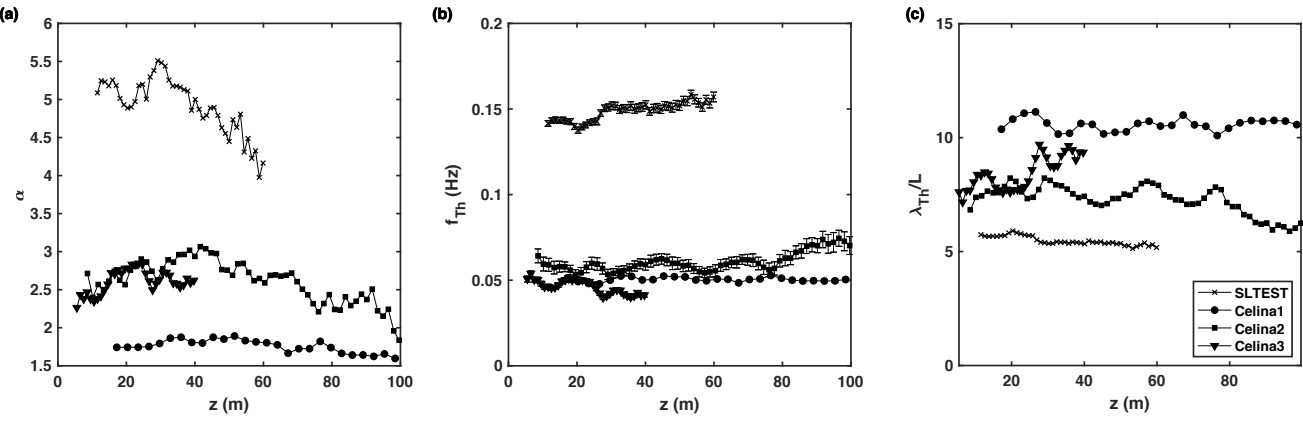

**Figure 15.** Low-pass filter parameters estimated for the various datasets: **(a)** filter order, $\alpha$; **(b)** cutoff frequency, $f_{Th}$; **(c)** cutoff wave length, $\lambda_{Th}$, normalized with the gate length, $L$.





# 6 Assessment of the correction procedure through synthetic velocity signals

To better understand the importance of the filter order, $\alpha$, and the cutoff frequency, $f_{Th}$ in Eq. 10, a turbulent synthetic signal is generated, by using a sampling frequency $f_S = 20$ Hz, from the inverse Fast Fourier Transform of a Kaimal-like spectrum (Eq. 1) with $A_f = 10$ m$^2$s$^{-1}$, $B_f = 20$ s, and providing random phases to the various spectral contributions. The generated time series consists of 36,000 samples evenly spaced in time.

This synthetic velocity signal is convoluted in the frequency domain with Eq. 10 where the order, $\alpha$, and the cutoff frequency,
$f_{Th}$, are varied within the intervals $(\alpha, f_{Th}/f_S) \in [2,5] \times [0.005, 0.25]$. The energy spectra obtained from the convolution of the synthetic signal with the various low-pass filters obtained by using the extrema of the above-mentioned ranges for $\alpha$ and $f_{Th}/f_S$ are reported in Fig. 16(a), while the respective low-pass filters are depicted in Fig. 16(b). It is evident that the largest deviation from the spectrum of the original synthetic signal is associated with the minimum value of the cutoff frequency (red and cyan lines in Figs. 16(a) and (b)). On the other hand, the filter order produces a secondary effect on the spectral damping,
namely a faster damping with increasing frequencies is observed with higher filter orders. To couple damping effects of both cutoff frequency and order of the low-pass filter, the percentage reduction of variance, which is calculated through Eq. 20, is reported in Fig. 16(c). This color map corroborates that the largest variability in the damping of the energy spectra occurs with variations of the cutoff frequency, while the gradient in the vertical direction, namely with variations of $\alpha$, is practically negligible. Specifically, for a cutoff frequency interval ranging from $0.5\%$ to $25\%$ of the sampling rate, the corresponding
energy reduction varies from $45\%$ down to $5\%$, which is in agreement with the experimental data reported in Fig. 14(b). Therefore, for the correction of the spectral damping of the LiDAR velocity signals, the main parameter to be tuned for the correction procedure is the cutoff frequency of the low-pass filter.

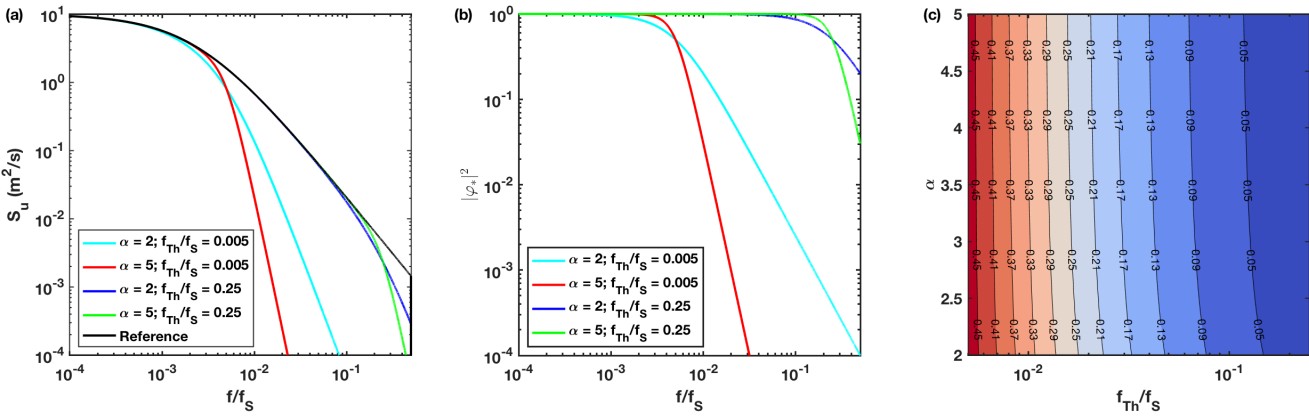

**Figure 16.** Convolution of a synthetic velocity signal: **(a)** reference Kaimal spectrum (solid black) used to generate the synthetic velocity signal and low-pass filtered spectra obtained by varying $\alpha$ and $f_{Th}$; **(b)** energy spectra of the used low-pass filters; **(c)** Percentage reduction of the velocity variance for different $\alpha$ and $f_{Th}$ values.





We now attempt to interpret the variability in the damping of the velocity variance as a function of height observed from the experimental datasets, as it has been reported in Fig. 14(b). For canonical boundary layer flows, we should expect a logarithmic

trend of the velocity variance as a function of the wall-normal distance (Townsend , 1976), as follows:

$$\frac{\sigma^2}{u_\tau^2} = H_1 - G_1 \log\left(\frac{z}{\delta}\right),\tag{21}$$

where $\delta$ is the surface layer height, while the parameters $H_1$ and $G_1$ might be dependent on the characteristics of the specific boundary layer flow under investigation. Previous field campaigns performed at the SLTEST site have quantified them as follows: $H_1 = 2.14$ and $G_1 = 1.33$ (Marusic et al. , 2013).

According to the Kaimal model, the velocity variance can be obtained by integrating Eq. 1 in the spectral domain, which leads to:

$$\frac{\sigma^2}{u_\tau^2} = \int_0^\infty \frac{A_f}{(1 + B_f f)^{5/3}} \, df = \frac{3}{2}\frac{A_f}{B_f},\tag{22}$$

where $A_f$ and $B_f$ are now functions of height.

To generate synthetic velocity signals throughout the boundary layer height similar to those observed from the experimental

datasets, the vertical profile of $B_f$ has been selected equal to that measured for the SLTEST dataset, which is related to the spectral peak reported with black empty circles in Fig. 11(b). Using the values of $H_1, G_1$ provided by Marusic et al. (2013), the logarithmic profile of variance is estimated from Eq. 21 over the height range $z/\delta \in [0.1 \quad 1]$. Finally, $A_f$ is calculated from Eq. 22.

The convolution of the synthetic velocity signals is performed by using $\alpha = 3$ and $f_{Th} \in [0.05, \quad 0.15]$ Hz. The sampling

rate is assumed equal to 20 Hz, while the total number of samples is $40,000$. The variance associated with the low-pass filtered velocity signals is reported as a function of height and for the various cutoff frequencies in Fig. 17(a). In agreement with the results reported in Fig. 16, a more severe damping is inferred to the synthetic velocity signals with reducing cutoff frequency. Furthermore, a lower cutoff frequency entails a more marked departure of the variance trend as a function of height from the expected logarithmic law, which also extends to higher heights.

By calculating the percentage reduction of the variance, $\epsilon$, through Eq. 20, it is observed that the damping in the variance due to the LiDAR measuring process typically decreases with height until a certain asymptotic error is achieved for heights between $0.3\delta$-$0.4\delta$. It is interesting that this feature singled out through the analysis of the synthetic velocity signals is very similar to that observed in Fig. 14(b) for the SLTEST dataset, for which the surface layer height, $\delta$ has been estimated between 60 m and 100 m from previous works (Hutchins & Marusic , 2007; Metzger et al. , 2007; Marusic and Hutchins , 2008). A similar

trend is also observed for the dataset Celina1 even though it exhibits a larger variance error (nearly $25\%$). For the remaining datasets, a roughly constant percentage error has been calculated, which might correspond to the region with asymptotic $\epsilon$ values observed in Fig. 17(b).





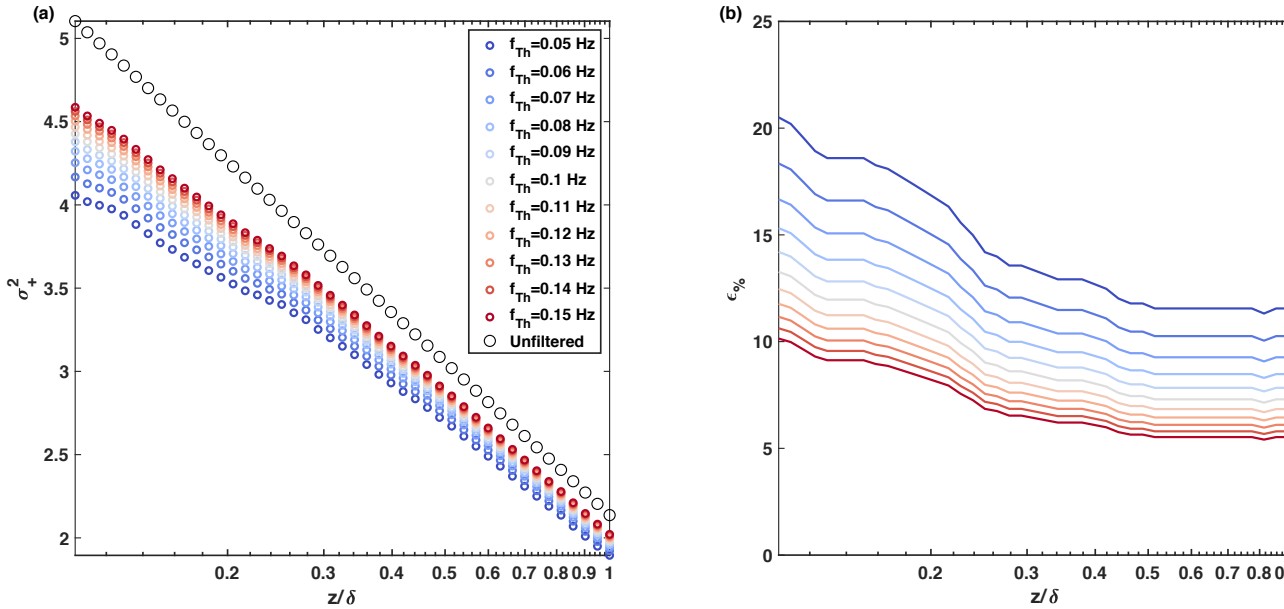

**Figure 17.** Assessment of the damping on LiDAR measurements through synthetic velocity signals: **(a)** unfiltered variance profile (empty black circles) and filtered variance profiles (blue to red empty circles); **(b)** Vertical profiles of normalized reduction of variance for various values of the cutoff frequency.

## 7 Conclusions

Wind LiDAR technology is gradually achieving compelling technical specifications, such as range gates smaller than 20 m and sampling frequencies higher than 1 Hz, which are instrumental to investigate atmospheric turbulence with length scales typical of the inertial sub-range. However, the emission of a laser pulse over the range gate to measure the radial velocity entails a spatial smoothing process leading to a damping on the measured variance of the velocity fluctuations. Existing models propose to correct the LiDAR measurements through a deconvolution procedure where the smoothing function represents the energy distribution of the laser pulse over the range gate. According to previous works, and also confirmed through this study, these de-convolution methods have limited accuracy in correcting the LiDAR velocity fluctuations.

In this work, we have proposed to correct the measured LiDAR velocity signals by inverting the effects of a low-pass filter representing the energy damping on the velocity fluctuations due to the LiDAR measuring process. The filter characteristics, namely order and cutoff frequency, are directly estimated from the LiDAR data under investigation. Specifically, the spectrum of the LiDAR velocity signal is fitted through the Kaimal spectral model for streamwise turbulence only for frequencies lower than a cutoff value for which the slope of the LiDAR velocity spectrum is observed to deviate from the expected $-5/3$ slope typical of the inertial sub-range. The ratio between the LiDAR and the Kaimal spectra is then fitted with the analytical expression of a low-pass filter to estimate order and cutoff frequency. The modeled low-pass filter is then reverted on the LiDAR



data to correct the LiDAR measurements and produce more accurate second order statistics and spectra of the streamwise wind velocity.

For this study, the proposed method for correction of the LiDAR data has been applied to datasets collected during three different field campaigns and for one dataset the procedure has been assessed against simultaneous and co-located sonic anemometer data. For this case, it has been shown that the proposed procedure allows to correct the second-order statistics of the LiDAR data to estimate a velocity variance comparable to that measured by a sonic anemometer.

This study has shown that it is challenging to determine a priori the cutoff frequency and order of the low-pass filter rep-
resenting the energy damping of the LiDAR measuring process for a given LiDAR system, LiDAR settings and atmospheric wind conditions. Therefore, we rather suggest to determine the parameters of the low-pass filter on a case-by-case basis by applying the proposed procedure on the LiDAR signal under investigation.

To better understand the role of the cutoff frequency and order of the low-pass filter representing the LiDAR energy damping, a further analysis has been conducted on synthetic turbulent velocity signals. This analysis has shown that the main parameter
for efficiently correct the LiDAR energy damping is the cutoff frequency of the low-pass filter, while the velocity statistics are weakly affected by the filter order. Subsequently, investigating effects of the LiDAR energy damping on synthetic turbulent velocity signals characterized by a logarithmic reduction of the variance with decreasing height from the ground, it has been shown that the LiDAR energy damping leads to a departure of the variance as a function of height from the expected logarithmic trend, while achieving an asymptotic percentage error with increasing heights. As expected, the energy damping is enhanced
with reducing cutoff frequency of the low-pass filter and it is extend to a higher altitudes.

Summarizing, the proposed procedure allows achieving efficient corrections of the second order statistics and spectra of the LiDAR measurements by inverting the effects of a low-pass filter, whose characteristics are directly estimated from the LiDAR data. However, further investigations with different LiDAR systems, LiDAR settings, such as range gate and accumulation time, and atmospheric wind conditions are needed to be able to predict a priori the correction parameters.

*Acknowledgements.* This work was supported by the National Science Foundation (NSF) CBET grant # 1705837, program manager Dr. Ronald Joslin.



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
