# Peer review of "Spectral correction of turbulent energy damping on wind LiDAR measurements due to spatial averaging"

_Atmospheric Measurement Techniques, 2020_

## Referee Comment (RC1) · Anonymous Referee #1 · 3 Jun 2020

**Spectral correction of turbulent energy damping on wind LiDAR measurements due to range-gate averaging**

Reviewer's comments

**1 General comment**

The manuscript "Spectral correction of turbulent energy damping on wind LiDAR measurements due to range-gate averaging" by Puccioni and Iungo deals with the problem of spatial filtering by the probe volume of pulsed Doppler wind lidar instruments. This spatial filtering challenges the proper characterization of turbulence. Therefore, they propose an empirical transfer function, which under certain conditions, can be used to empirically correct the filtering effect. This transfer function is fitted to the ratio between the estimated power spectral density (PSD) of the along-beam velocity component and an empirical spectral model. This empirical model is based on the Blunt model (Olesen et al., 1984) and is fitted to the low-frequency range of the estimated velocity spectrum.

The solution proposed by Puccioni and Iungo is practical and simple, which is appreciated. The dataset is not novel but this is not so important here. In this regards, the paper is within the scope of AMT. The language is fluent but sometimes unprecise or unclear. While the conclusions of the paper support the proposed method and the overall content is clear, some major questions are raised by the manuscript, which should be addressed:

- I have the impression that a significant portion of the manuscript is filling. The size of the manuscript could be reduced by 40 % without affecting its core message. The value of a paper is not defined by its length, fortunately. The filling is sometimes counter-productive as shown in section 6, where the authors use synthetic turbulence generation to work exclusively in the frequency domain: there is no need to generate any turbulent field in the time-domain if the calculations are conducted on the velocity spectra only. This section deals with an interesting topic, which is the dependency of the spatial averaging on the mean wind speed and the variance of the velocity component. However, unnecessary steps are used to reach the conclusion, which erode the analysis.

- The data processing is not always clear. Also, data are sometimes over-processed. The main fitting algorithm is likely applicable only if the spectral peak is not affected by spatial filtering. If the spectral peak is filtered out, the peak frequency will become "corrupted". This limitation is not clearly highlighted in the manuscript.

- References to the existing scientific literature can be inaccurate or misleading. The number of self-citations in the manuscript is equal to almost one-third of the total number of references, which might be a little too high.

These three points are described more in details in section 2. Puccioni and Iungo deal with a challenging issue and their simple approach is, therefore, welcome. For these reasons, I recommend a major revision of the manuscript.

**2 Specific comments**

**Point 1**

Title: The term "range-gate averaging" may be criticized because the range gate does not necessarily refer to the probe volume length. I suggest using the term "spatial averaging" or "volume averaging" as a safe alternative.

**Point 2**

The first paragraph of the introduction reviews previous turbulence measurements in the atmosphere by Doppler wind lidar instruments. The majority of these references is inadequate:

- It is unclear how the work by Trukenmüller et al. (2004), Horányi et al. (2015) or Schepers et al. (2012) are related to Doppler Wind lidar measurements. I suggest removing these references.

- The works by Calhoun et al. (2006), Vanderwende et al. (2015) and El-Asha et al. (2017) are interesting but they are not about turbulence measurements. Their focus was on the mean wind speed only. I suggest removing these references.

- The reference to Grubišić et al. (2008) may not be appropriate because the lidars were not used to investigate turbulence characteristics. If the authors believe that a similar study must be included, the work by Spuler and Mayor (2005) might be more relevant. Note that Spuler and Mayor only collected snapshots of coherence structures, which may not be considered as "turbulence characterization" but rather "flow visualization".

- The reference to Fernando et al. (2019) may be replaced by the reference to Bodini et al. (2017) since the method used by Fernando et al. to study the turbulence dissipation rate is taken from Bodini et al.

- The reference to George and Yang (2012) may be removed because it is a review paper on vortices detection by various instruments. They did not focus on turbulence characterization and did not show any results from Doppler wind lidar measurements.

- Only self-references are used to illustrate turbulence measurements by lidars in the field of wind energy. In addition, the same results are sometimes cited multiple times because they are included in different similar papers. I recommend choosing only one of these papers and to not use self-references only.

The authors can find hereafter some of additional studies by (scanning) Doppler wind lidar instruments focusing of turbulence characterization: Lothon et al. (2006, 2009); Newsom et al. (2008); Angelou et al. (2012); Sathe and Mann (2012a); Branlard et al. (2013); Cheynet et al. (2016); Wang et al. (2016); van Dooren et al. (2017); Kumer et al. (2017); Held and Mann (2018); Peña and Mann (2019); Wildmann et al. (2019); Mauder et al. (2020). I also invite the authors to search for additional studies available in the scientific literature.

**Point 3**

Section 1: Some lines mentioning that the paper focuses on scanning pulsed Doppler wind lidar and not continuous-wave lidars or wind profilers may be necessary for the sake of clarity.

**Point 4**

Line 27-28: The sentence "Turbulence statistics of the wind velocity field can be retrieved through fixed scans while providing a spectral characterization of the inertial sub-layer" is only partly true. If the probe volume is larger than 50 m, there exist situations where spatial filtering can affect the entire inertial subrange, preventing the detailed characterization of turbulence.

**Point 5**

Line 29: The reference to the detection of very large coherence structures is a little strange here because it does not imply the possibility to establish turbulence statistics from them. In particular, the experiment by Calaf et al was done without knowing precisely the wind direction as they had no access to wind vanes or anemometers. Besides, the scientific literature contains many more examples of turbulence characteristics retrieved from fixed line-of-sight scans.

**Point 6**

Line 32: The reference to Mann et al. (2009) is only partly true: The actually estimated the auto and cross-spectral densities for the three velocity components, which is more advanced than the turbulent momentum flux.

**Point 7**

Line 34: A probe volume below 20 m is not so common for commercially available pulsed Doppler wind lidar. Maybe some comments can be written here.

**Point 8**

Line 42-43: The influence of the misalignment between the wind direction and laser beam could also be mentioned as an additional effect on the spatial filtering by the probe volume (see e.g. Held

and Mann ([2018])).

**Point 9**

Line 49: In Cheynet et al. ([2017]), the probe volume length was 75 m and the range gate length was 100 m. The probe length of 100 m mentioned in their study was used as an example to illustrate the spatial filtering.

**Point 10**

In section 2, equation (1) is not necessary for the paper. Since the spatial filtering is a function of the wavenumber, using $f$ (in Hertz) instead of $n = fz/U$ is not desirable. Therefore, the study can be simplified by considering only equation 2.

**Point 11**

The Kaimal model and Simiu-Scanlan models are particular cases of Equation 2. Note that in Kaimal et al. ([1972]), An = 105 and Bn = 33 but in Kaimal and Finnigan ([1994]), An = 102 and Bn = 33. Equation (2) with unspecified $A_n$ and $B_n$ values should be referred to as the blunt model (Olesen et al., [1984]) instead of Kaimal model. The reference to ESDU is incorrect here. The ESDU standard is using a modified von Karman model, which has a form different from Equation 2. Therefore, I suggest removing the reference to ESDU.

**Point 12**

The algorithm in Figure 1 is interesting but also perfectible. It does not clearly show why the iterative procure is necessary and this should be explained in a pedagogical way. For example, it could be stated that there is no need to have an iterative procedure if $f_{Th,0}$ is equal or lower than 0.01 Hz.

There is an argument in favour of the iterative procedure that is not clearly stated in section 2: Choosing a value of $f_{Th,0}$ too low will result in a poor fit of the velocity spectrum because the number of data points will be reduced. In addition, these points are associated with larger uncertainties than at higher frequencies. At the same time, if $f_{Th,0}$ is larger than the cut-off frequency, the fitting will be significantly affected by the spatial filtering.

There is also a potential limit for the application of this algorithm that was not clearly shown in the manuscript: the spectral correction may fail if the spectral peak is affected by the spatial filtering. Therefore the proposed method might only be adequate for probe volume of 50 m or lower. That is an issue that deserves further discussion.

**Point 13**

Equation 10: The spatial filtering is a function of the wavenumber rather than the frequency.

Therefore, I think that fitting a modified version of Eq. 10, where the frequency is replaced by the wavenumber, may be more appropriate than the original version of Eq. 10.

**Point 14**

Section 2: is the fitting algorithm a least-square fit?

**Point 15**

Line 145: Was the lidar azimuth set manually as equal to the mean wind direction or was it an automated procedure?

**Point 16**

Line 148: Maybe it can be explained why the sampling frequency was varying between 0.5 Hz and 3.3 Hz?

**Point 17**

Figure 2: The topography is a little difficult to see. Maybe you can use a digital terrain model?

**Point 18**

Line 163: Since the Obukhov length is calculated, I suggest replacing "static atmospheric stability" by "dynamic atmospheric stability" or simply "atmospheric stability".

**Point 19**

Line 178: I do not understand the link between the sentence and the reference to Hutchins et al. (2012). Maybe this reference is not necessary?

**Point 20**

Lines 182-191: These lines could be summarized into a single sentence: "The second-order stationarity is assessed using a moving standard deviation with a window length of 5 min and zero overlapping". The reference to Liu et al is not adequate since they did not invent the concept of moving standard deviation. Besides, I would recommend using overlapping windows for a more robust assessment of the flow stationarity. In Matlab, the function "movstd" can be used for this purpose.

**Point 21**

The test of the second-order stationarity is a nice addition by the authors. I would also recommend a test for the first-order stationarity using a moving mean function.

**Point 22**

Line 190: Is there any reason for choosing 40 % for the maximal IST value?

**Point 23**

Line 192: I am not sure I understand the "gradient-based" procedure to remove outliers. Maybe one sentence can be written to make it clearer?

**Point 24**

Line 194: There is no official Matlab function "inpaint". However, there exists the function "inpaint_nans" by D'Errico (2004), which is well respected. Is this the function that you were using? If yes, a reference to D'Errico (2004) can be used.

**Point 25**

Eq. 17: This equation is only valid for the mean wind speed. If the variance is computed, substantial errors may arise (see eq. 2 in Sathe and Mann (2012b), which is also valid for a LOS scan mode). Some comments are expected here.

**Point 26**

Line 228-229: The Savitzky-Golay smoothing filter was not designed by Balasubramaniam (2005) but by Savitzky and Golay (1964). I do not recommend using this filter to smooth the power spectral density (PSD) as it also distorts the low-frequency range of the spectrum. The goal should be to smooth only the high-frequency range, which includes enough data point. I advise to simply bin-average your data over bins that are uniformly spaced in the logarithmic space. This way, the fitting algorithm will also be improved. Alternatively, you can estimate the PSD using auto-regressive methods, which can produce fairly smooth PSD estimates if the random process of interest is broad-banded, which is the case here.

**Point 27**

The method to estimate the velocity spectrum should be explicitly stated. Since a fitting to the velocity spectrum is done, it is important to know if the PSD estimate is reliable or not. I do not recommend using the periodogram method. Alternatives approaches are the modified periodogram method (Welch, 1967) or the multitaper method (Thomson, 1982), both available in Matlab.

**Point 28**

Figure 5: The high-frequency range of the downsampled velocity spectrum seems to be slightly too high. This might be due to the presence of aliasing if the time series were downsampled without filtering. Decimating the original time series with a FIR filter and an order equal to at least 10 is recommended (cf. the Matlab function 'decimate'). This should lead to a better agreement between the velocity spectra of the lidar data and the sonic data.

**Point 29**

Line 263-264: The small difference between the difference range gate length is unlikely to explain the difference between the different filter functions. One possible reason for the discrepancies is the presence of measurement noise, which increases with the frequency and which is accounted for in the empirical filter function used in the manuscript but not modelled in eqs. 6-7.

**Point 30**

Figure 7 and the associated discussion do not seem to be a vital piece of information here. Firstly, the use of subsamples with an averaging time of $25\,\mathrm{s}$ could be criticized as it only includes a small portion of the turbulence spectrum. Therefore, the influence of the correction method on the variance estimate becomes exaggerated, which is not desirable in relation to algorithm validation. Secondly, only one time series is included, which limits the conclusions that can be drawn from this figure. It may be a careful choice to remove this figure and the corresponding paragraph.

**Point 31**

Line 284-285: When applying a high-pass filter, the reader needs to know the exact value of the cut-off frequency, the order of the filter and what type of filter is applied. Therefore, mentioning a cut-off frequency of the order of $10^{-3}\,\mathrm{Hz}$ is not sufficient. Velocity fluctuations around $1 \times 10^{-3}\,\mathrm{Hz}$ may still be representative of micro-scale turbulence. The use of the high-pass filter will increase the influence of the correction algorithm on the estimation of turbulence characteristics. This can be mentioned and/or quantified.

**Point 32**

Figure 8 includes two subfigures that can be merged into a single one by using a 3-variable scatter plot, where the colour of the markers reflects the variance. Nevertheless, I am not sure that figure 8 is vital to the paper.

**Point 33**

Reference to Guala et al. (2006) is inadequate. They studied turbulence in pipe flows whereas the paper discusses turbulence in the atmospheric boundary layer. A more appropriate reference would be Counihan (1970).

**Point 34**

Figure 9 can be reduced to a single panel. I think showing the pre-multiplied spectrum $fS_u$ as a function of the wavenumber $k$ is good enough.

**Point 35**

Line 320: "selected dataset" is not specific enough. Maybe you can mention which data from which campaigns?

**Point 36**

Figure 11 looks nice but it is not necessary to the paper. Firstly because we cannot see a clear difference between the left and right panel and secondly because it does not bring particularly useful information. I suggest removing this figure for the sake of brevity.

**Point 37**

Figure 12 is not necessary either to the paper. The textual description was already enough. I think this figure can be removed.

**Point 38**

Line 330: I am not sure what you mean by a quasi self-similar behaviour. Figure 13 is showing (normalized) transfer functions. I think it may be wise to keep the description as simple as possible.

**Point 39**

Figure 15 could be replaced by a simple table showing the median value and the interquartile range, for example.

**Point 40**

Section 6 could be reformulated into one or two paragraphs. Firstly, the use of synthetic wind field is not useful here, as calculations are conducted in the frequency domain only. Secondly, the computation of the profiles of the mean wind speed and variance as well as the associated discussion is unnecessary. The right panel of Fig 17 is interesting but does not shows that the error $\varepsilon$ could be expressed as a function of the mean wind speed and variance of the velocity only.

I suggest shortening in section 6. It is possible to show the dependency of $\varepsilon$ on the mean wind speed and the variance of the velocity as a contour map. Given a reference mean wind speed at a reference height, a logarithmic profile with and a given roughness length and the Kaimal spectrum, constructing such a map is straightforward. To change the mean wind speed parameter, you simply need to change the reference mean wind speed value. To change the variance of the along-wind component, you can simply change the roughness length.

**Point 41**

The conclusion should be reformulated following the previous comments

**3 Technical corrections**

**Point 1**

Lines 1-3 (abstract): Maybe you should mention that you are talking about "pulsed" lidar systems. Continuous-wave Doppler wind lidars can measure the flow within a volume much lower than 20 m and a sampling frequency of several hundreds of Hertz.

**Point 2**

Line 2 (abstract): a sampling frequency of the order of 10 Hz is mentioned. Do you mean 1 Hz, as written in the manuscript?

**Point 3**

Line 3 (abstract): the expression "back-scattered laser beam" may not be correct. Do you mean "backscattered light" or "backscattered signal"?

**Point 4**

Lines 9 (abstract): I suggest replacing "estimated directly from the LiDAR measurement" by a more accurate term: "estimated directly from the power spectral densities of the along-beam velocity component".

**Point 5**

Line 34: The sentence "probe volumes, denoted as range gates" can be misunderstood by the reader and should be reformulated. The range gate length is different from the probe volume. For example, a range gate length shorter than the probe volume implies that the probe volumes are overlapping.

**Point 6**

Line 36: The syntax of the sentence "by means of a laser beam, which is back-scattered" is a little strange. I would write that the light is backscattered but not that the "laser beam" is backscattered.

**Point 7**

Line 38: " from the Doppler shift **on** the back-scattered signal" should be "from the Doppler shift **of** the back-scattered signal"

**Point 8**

Line 38: "like those used for the present work" should be "like the one used in the present work".

**Point 9**

The tilde symbol in Equation 10 and equation 11 may be explicitly defined for the sake of clarity.

**Point 10**

Line 156: "of the used LiDARs" may be written "of the LiDARs used" instead.

**Point 11**

Line 175-176: "For the SLTEST and Celina campaigns [...]through DBS or VAD scans" seems to be a repetition of the same information mentioned earlier in the manuscript. Maybe this sentence can be removed.

**Point 12**

Line 192: Consider replacing present tense by past tense when describing the data processing.

**Point 13**

Line 251: The "spectrum of the lidar signal" should be replaced with "spectrum of the LOS velocity"

**Point 14**

Line 276: You may replace "linear regression analysis" by "comparison", which is much simpler

**Point 15**

Line 287: The term "first and second-order statistics" can be replaced by "mean value and variance", which are the quantities you study in the paper.

**Point 16**

Line 307: The sentence "fitting of the LiDAR spectra" should be replaced with "fitting of the Blunt model to the along-beam velocity spectra"

**Point 17**

Line 324: "highest LiDAR gate" should be replaced by "highest LiDAR range gate" or "LiDAR range gate furthest from the instrument".

**Point 18**

Line 336: " [...] always underestimate [...]" Do you mean "overestimate" ?

**Point 19**

Figure 13: The term "convolution function" sounds strange. I would call it "transfer function" as it is the case in the field of signal processing

**Point 20**

Line 359: Convolution in the time domain is multiplication in the frequency domain. Therefore writing that the "synthetic velocity signal is convoluted in the frequency domain" is unclear. I think it may be simpler to write that the velocity spectrum is multiplied with the transfer function modelling the spatial averaging.

**References**

Olesen, H.R., Larsen, S.E., Højstrup, J.. Modelling velocity spectra in the lower part of the planetary boundary layer. Boundary-Layer Meteorology 1984;29(3):285–312.

Trukenmüller, A., Grawe, D., Schlünzen, K.H.. A model system for the assessment of ambient air quality conforming to ec directives. Meteorologische Zeitschrift 2004;13(5):387–394.

Horányi, A., Cardinali, C., Rennie, M., Isaksen, L.. The assimilation of horizontal line-of-sight wind information into the ecmwf data assimilation and forecasting system. part i: The assessment of wind impact. Quarterly Journal of the Royal Meteorological Society 2015;141(689):1223–1232.

Schepers, J., Obdam, T., Prospathopoulos, J.. Analysis of wake measurements from the ecn wind turbine test site wieringermeer, ewtw. Wind Energy 2012;15(4):575–591.

Calhoun, R., Heap, R., Princevac, M., Newsom, R., Fernando, H., Ligon, D.. Virtual towers using coherent doppler lidar during the joint urban 2003 dispersion experiment. Journal of Applied meteorology and climatology 2006;45(8):1116–1126.

Vanderwende, B.J., Lundquist, J.K., Rhodes, M.E., Takle, E.S., Irvin, S.L.. Observing and simulating the summertime low-level jet in central iowa. Monthly Weather Review 2015;143(6):2319–2336.

El-Asha, S., Zhan, L., Iungo, G.V.. Quantification of power losses due to wind turbine wake interactions through scada, meteorological and wind lidar data. Wind Energy 2017;20(11):1823–1839.

Grubišić, V., Doyle, J.D., Kuettner, J., Mobbs, S., Smith, R.B., Whiteman, C.D., et al. The terrain-induced rotor experiment: A field campaign overview including observational highlights. Bulletin of the American Meteorological Society 2008;89(10):1513–1534.

Spuler, S.M., Mayor, S.D.. Scanning eye-safe elastic backscatter lidar at 1.54 $\mu$ m. Journal of Atmospheric and Oceanic Technology 2005;22(6):696–703.

Fernando, H., Mann, J., Palma, J., Lundquist, J.K., Barthelmie, R.J., Belo-Pereira, M., et al. The perdigao: Peering into microscale details of mountain winds. Bulletin of the American Meteorological Society 2019;100(5):799–819.

Bodini, N., Zardi, D., Lundquist, J.K.. Three-dimensional structure of wind turbine wakes as measured by scanning lidar. Atmospheric Measurement Techniques 2017;10(8).

George, R., Yang, J.J.. A survey for methods of detecting aircraft vortices. In: ASME 2012 International Design Engineering Technical Conferences and Computers and Information in Engineering Conference. American Society of Mechanical Engineers Digital Collection; 2012, p. 41–50.

Lothon, M., Lenschow, D.H., Mayor, S.D.. Coherence and scale of vertical velocity in the convective boundary layer from a doppler lidar. Boundary-layer meteorology 2006;121(3):521–536.

Lothon, M., Lenschow, D.H., Mayor, S.D.. Doppler lidar measurements of vertical velocity spectra in the convective planetary boundary layer. Boundary-layer meteorology 2009;132(2):205–226.

Newsom, R., Calhoun, R., Ligon, D., Allwine, J.. Linearly organized turbulence structures observed over a suburban area by dual-doppler lidar. Boundary-layer meteorology 2008;127(1):111–130.

Angelou, N., Mann, J., Sjöholm, M., Courtney, M.. Direct measurement of the spectral transfer function of a laser based anemometer. Review of scientific instruments 2012;83(3):033111.

Sathe, A., Mann, J.. Measurement of turbulence spectra using scanning pulsed wind lidars. Journal of Geophysical Research: Atmospheres 2012a;117(D1).

Branlard, E., Pedersen, A.T., Mann, J., Angelou, N., Fischer, A., Mikkelsen, T., et al. Retrieving wind statistics from average spectrum of continuous-wave lidar. Atmospheric Measurement Techniques 2013;6:1673–1683.

Cheynet, E., Jakobsen, J.B., Snæbjörnsson, J., Mikkelsen, T., Sjöholm, M., Mann, J., et al. Application of short-range dual-doppler lidars to evaluate the coherence of turbulence. Experiments in Fluids 2016;57(12):184.

Wang, H., Barthelmie, R.J., Doubrawa, P., Pryor, S.C.. Errors in radial velocity variance from doppler wind lidar. Atmospheric Measurement Techniques (Online) 2016;9(8).

van Dooren, M.F., Campagnolo, F., Sjöholm, M., Angelou, N., Mikkelsen, T.K., Kühn, M.. Demonstration and uncertainty analysis of synchronised scanning lidar measurements of 2-d velocity fields in a boundary-layer wind tunnel. Wind Energy Science 2017;2:329–341.

Kumer, V.M., Reuder, J., Oftedal Eikill, R.. Characterization of turbulence in wind turbine wakes under different stability conditions from static doppler lidar measurements. Remote Sensing 2017;9(3):242.

Held, D.P., Mann, J.. Comparison of methods to derive radial wind speed from a continuous-wave coherent lidar doppler spectrum. Atmospheric Measurement Techniques 2018;11(11).

Peña, A., Mann, J.. Turbulence measurements with dual-doppler scanning lidars. Remote Sensing 2019;11(20):2444.

Wildmann, N., Bodini, N., Lundquist, J.K., Bariteau, L., Wagner, J.. Estimation of turbulence dissipation rate from doppler wind lidars and in-situ instrumentation in the perdigão 2017 campaign. Atmospheric Measurement Techniques 2019;2019(12):6401–6423.

Mauder, M., Eggert, M., Gutsmuths, C., Oertel, S., Wilhelm, P., Voelksch, I., et al. Comparison of turbulence measurements by a csat3b sonic anemometer and a high-resolution bistatic doppler lidar. Atmospheric Measurement Techniques 2020;13(2).

Mann, J., Cariou, J.P., Courtney, M.S., Parmentier, R., Mikkelsen, T., Wagner, R., et al. Comparison of 3d turbulence measurements using three staring wind lidars and a sonic anemometer. Meteorologische Zeitschrift 2009;18(2):135–140.

Cheynet, E., Jakobsen, J.B., Snæbjörnsson, J., Mann, J., Courtney, M., Lea, G., et al. Measurements of surface-layer turbulence in a wide norwegian fjord using synchronized long-range doppler wind lidars. Remote Sensing 2017;9(10):977.

Kaimal, J.C., Wyngaard, J., Izumi, Y., Coté, O.. Spectral characteristics of surface-layer turbulence. Quarterly Journal of the Royal Meteorological Society 1972;98(417):563–589.

Kaimal, J.C., Finnigan, J.J.. Atmospheric boundary layer flows: their structure and measurement. Oxford university press; 1994.

Hutchins, N., Chauhan, K., Marusic, I., Monty, J., Klewicki, J.. Towards reconciling the large-scale structure of turbulent boundary layers in the atmosphere and laboratory. Boundary-layer meteorology 2012;145(2):273–306.

D'Errico, J.. Inpaint nans. MATLAB Central File Exchange 2004;.

Sathe, A., Mann, J.. Turbulence measurements using six lidar beams. In: 16th International Symposium for the Advancement of Boundary-Layer Remote Sensing. Steering Committee of the 16th International Symposium for the Advancement . . . ; 2012b, p. 302–305.

Balasubramaniam, B.J.. Nature of turbulence in wall-bounded flows. University of Illinois at Urbana-Champaign; 2005.

Savitzky, A., Golay, M.J.. Smoothing and differentiation of data by simplified least squares procedures. Analytical chemistry 1964;36(8):1627–1639.

Welch, P.. The use of fast fourier transform for the estimation of power spectra: a method based on time averaging over short, modified periodograms. IEEE Transactions on audio and electroacoustics 1967;15(2):70–73.

Thomson, D.J.. Spectrum estimation and harmonic analysis. Proceedings of the IEEE 1982;70(9):1055–1096.

Guala, M., Hommema, S., Adrian, R.. Large-scale and very-large-scale motions in turbulent pipe flow. Journal of Fluid Mechanics 2006;554:521–542.

Counihan, J.. Further measurements in a simulated atmospheric boundary layer. Atmospheric Environment (1967) 1970;4(3):259–275.

---

## Author Comment (AC1) · 20 Jul 2020

**Reply to the comments provided by the Anonymous Referee #1 on the manuscript amt-2020-27 entitled "Spectral Correction of turbulent energy damping on wind LiDAR measurements due to spatial averaging", by M. Puccioni and G. V. Iungo**

The authors are greatly thankful to the Reviewer for the thorough review and insightful comments. Our replies are reported in the following. References to pages and lines are based on the revised marked-up manuscript.

**General comment**

- *The manuscript "Spectral correction of turbulent energy damping on wind LiDAR measurements due to range-gate averaging" by Puccioni and Iungo deals with the problem of spatial filtering by the probe volume of pulsed Doppler wind lidar instruments. This spatial filtering challenges the proper characterization of turbulence. Therefore, they propose an empirical transfer function, which under certain conditions, can be used to empirically correct the filtering effect. This transfer function is fitted to the ratio between the estimated power spectral density (PSD) of the along-beam velocity component and an empirical spectral model. This empirical model is based on the Blunt model (Olesen et al., 1984) and is fitted to the low-frequency range of the estimated velocity spectrum. The solution proposed by Puccioni and Iungo is practical and simple, which is appreciated. The dataset is not novel but this is not so important here. In this regard, the paper is within the scope of AMT. The language is fluent but sometimes unprecise or unclear. While the conclusions of the paper support the proposed method and the overall content is clear...*

R: We thank the Reviewer for the positive feedback on our research strategy and the results achieved. It has been instrumental to leverage various LiDAR datasets, which have been collected by our group in collaboration with other colleagues, to prove the general applicability of the proposed model. Writing has been improved throughout the manuscript.

- *I have the impression that a significant portion of the manuscript is filling. The size of the manuscript could be reduced by 40 % without affecting its core message. The value of a paper is not defined by its length, fortunately. The filling is sometimes counter-productive as shown in section 6, where the authors use synthetic turbulence generation to work exclusively in the frequency domain: there is no need to generate any turbulent field in the time-domain if the calculations are conducted on the velocity spectra only. This section deals with an interesting topic, which is the dependency of the spatial averaging on the mean wind speed and the variance of the velocity component. However, unnecessary steps are used to reach the conclusion, which erode the analysis.*

R: We are research enthusiasts and very meticulous in the execution of our projects. If sometimes, unfortunately, we end up writing lengthy manuscripts is definitely not connected with the need of filling a document, which is clearly not needed considering the length of the text, rather prove the research assumptions and corroborate our results. We have significantly shortened the manuscript and sharpened its focus. Sect. 6 has been significantly revised, and only the part related to the variability of the energy damping with wind speed, standard deviation, and sampling height is kept.

- *The data processing is not always clear. Also, data are sometimes over-processed. The main fitting algorithm is likely applicable only if the spectral peak is not affected by spatial filtering. If the spectral peak is filtered out, the peak frequency will become "corrupted". This limitation is not clearly highlighted in the manuscript.*

R: We definitely agree with the Reviewer's comment and the mentioned constraint, namely ensuring that the spectral peak in not corrupted during the post-processing, has been always verified for our data analysis. In the revised version of the manuscript, we have highlighted the importance of verifying that the energy content in the proximity of the spectral peak is not altered by the post-processing procedure. In the manuscript at line 147, it is now reported: "If during the iterative process, $k_{Th}$ achieves a value equal or smaller than that corresponding to the spectral peak, $k_p$, then the procedure is arrested and a warning is dispatched indicating that the correction procedure was not successful".

- *References to the existing scientific literature can be inaccurate or misleading. The number of self-citations in the manuscript is equal to almost one-third of the total number of references, which might be a little too high.*

R: As recommended by the Reviewer, we have significantly shortened our reference list.

**Specific comments**

**Point 1**

*The term "range-gate averaging" may be criticized because the range does not necessarily refer to the probe volume length. I suggest using the term "spatial averaging" or "volume averaging" as a safe alternative.*

R: During the preparation of this manuscript, we were skeptical in using the term spatial averaging because different spatial-averaging processes may occur when operating scanning wind LiDARs, such as by varying continuously the scanning head in the azimuthal direction for VAD and PPI scans, or the elevation angle for RHI scans. Throughout the manuscript, the smoothing process under investigation is now referred to as spatial averaging, consistently with previous works, e.g. Frehlich et al. (1998) and Sjöholm et al. (2009).

**Point 2**

*The first paragraph of the introduction reviews previous turbulence measurements in the atmosphere by Doppler wind lidar instruments. The majority of these references is inadequate:*
- *It is unclear how the work by Trukenmüller et al. (2004), Horányi et al. (2015) or Schepers et al. (2012) are related to Doppler Wind lidar measurements. I suggest removing these references.*
- *The works by Calhoun et al. (2006), Vanderwende et al. (2015) and El-Asha et al. (2017) are interesting but they are not about turbulence measurements. Their focus was on the mean wind speed only. I suggest removing these references.*
- *The reference to Grubišic et al. (2008) may not be appropriate because the lidars were not used to investigate turbulence characteristics. If the authors believe that a similar study must be included, the work by Spuler and Mayor (2005) might be more relevant. Note that Spuler and Mayor only collected snapshots of coherence structures, which may not be considered as "turbulence characterization" but rather "flow visualization".*

- *The reference to Fernando et al. (2019) may be replaced by the reference to Bodini et al. (2017) since the method used by Fernando et al. to study the turbulence dissipation rate is taken from Bodini et al.*
- *The reference to George and Yang (2012) may be removed because it is a review paper on vortices detection by various instruments. They did not focus on turbulence characterization and did not show any results from Doppler wind lidar measurements.*
- *Only self-references are used to illustrate turbulence measurements by lidars in the field of wind energy. In addition, the same results are sometimes cited multiple times because they are included in different similar papers. I recommend choosing only one of these papers and to not use self-references only.*

R: We have revised the mentioned references according with the comments of the Reviewer.

**Point 3**

*Section 1: Some lines mentioning that the paper focuses on scanning pulsed Doppler wind lidar and not continuous-wave lidars or wind profilers may be necessary for the sake of clarity.*

R: We definitely agree with this comment, indeed our discussion focuses completely on pulsed wind LiDARs. For instance, at line 45 it is reported: "A pulsed Doppler wind LiDAR, like those used for the present work…".

**Point 4**

*Line 27-28: The sentence "Turbulence statistics of the wind velocity field can be retrieved through fixed scans while providing a spectral characterization of the inertial sub-layer" is only partly true. If the probe volume is larger than 50 m, there exist situations where spatial filtering can affect the entire inertial subrange, preventing the detailed characterization of turbulence.*

R: That sentence has been revised as (line 30): "Provided the use of a probe length, $l$, sufficiently small to probe the inertial sublayer at a height from the ground $z$, e.g. $l < 2\pi z$ according to Banerjee et al. (2015), turbulence statistics of the wind velocity field can be retrieved through fixed scans, while providing a spectral characterization of the inertial sub-layer (Iungo et al., 2013)".

**Point 5**

*Line 29: The reference to the detection of very large coherence structures is a little strange here because it does not imply the possibility to establish turbulence statistics from them. In particular, the experiment by Calaf et al. was done without knowing precisely the wind direction as they had no access to wind vanes or anemometers. Besides, the scientific literature contains many more examples of turbulence characteristics retrieved from fixed line-of-sight scans.*

R: We thank the Reviewer for this comment. The reference to the paper Calaf et al. (2013) and the related text have been removed.

**Point 6**

*Line 32: The reference to Mann et al. (2009) is only partly true: They actually estimated the auto and cross-spectral densities for the three velocity components, which is more advanced than the turbulent momentum flux.*

R: The Reviewer is right. At line 35, it is now reported: "In Mann *et al.* (2009), auto- and cross-spectral densities for the three velocity components were estimated through multiple scanning-LiDAR measurements".

**Point 7**

*Line 34: A probe volume below 20 m is not so common for commercially available pulsed Doppler wind lidar. Maybe some comments can be written here.*

R: Only LiDAR units for research on ABL turbulence may provide capabilities to perform measurements with very short-range gates, e.g. 20 m. At line 38, it is now reported: "… wind LiDARs tailored for investigations on atmospheric-turbulence currently provide probe volumes smaller than 20 m…".

**Point 8**

*Line 42-43: The influence of the misalignment between the wind direction and laser beam could also be mentioned as an additional effect on the spatial filtering by the probe volume (see e.g. Held and Mann (2018)).*

R: This is correct. Indeed, at line 130, it is reported: "… features of the low-pass filter and, thus, of the LiDAR measuring process, are functions of …relative angle between wind direction and azimuth angle of the laser beam …".

**Point 9**

*Line 49: In Cheynet et al. (2017), the probe volume length was 75 m and the range gate length was 100 m. The probe length of 100 m mentioned in their study was used as an example to illustrate the spatial filtering.*

R: At line 53, that statement is now revised as: "For single-point measurements performed with a Windcube 200S LiDAR and azimuthal angle of the laser beam set equal to the mean wind direction, a variance reduction of 8% was predicted for a gate length of 25 m, while it was increased up to 20% for a gate length of 100 m (Cheynet et al. , 2017)."

**Point 10**

*In section 2, equation (1) is not necessary for the paper. Since the spatial filtering is a function of the wavenumber, using f (in Hertz) instead of n = f z/U is not desirable. Therefore, the study can be simplified by considering only equation 2.*

R: We agree that providing both formulas might be redundant. The only equation with the reduced frequency, *n*, is now reported.

**Point 11**

*The Kaimal model and Simiu-Scanlan models are particular cases of Equation 2. Note that in Kaimal et al. (1972), An = 105 and Bn = 33 but in Kaimal and Finnigan (1994), An = 102 and Bn = 33. Equation (2) with unspecified An and Bn values should be referred to as the blunt model (Olesen et al., 1984) instead of Kaimal model. The reference to ESDU is incorrect here. The ESDU standard is using a modified von Karman model, which has a form different from Equation 2. Therefore, I suggest removing the reference to ESDU.*

R: We added in the manuscript that in Olsen *et al., 1984* the used spectral model is referred to as blunt model. However, even in that paper, it is reported that the spectral model was already used in Kaimal *et al.* 1972, for unstable conditions in Kaimal *et al.* 1976, Panofsky 1978, Højstrup 1981 and 1982, while for stable conditions in Kaimal 1973 and Caughey 1977. More recent papers refer to this spectral model as Kaimal model, see e.g. Risan *et al.* 2018, Worsnop *et al.* 2017 and even in the IEC standards for wind energy (International Electrotechnical Commission (2007) IEC 61400-1: Wind turbines—part 1: design requirements. 3rd edn). In the text at line 91, it is now

reported: "The spectral model of Eq. 1 is typically referred to as blunt model (Olesen *et al.*, 1984) or Kaimal model (Kaimal *et al.*, 1972; IEC, 2007; Worsnop *et al.*, 2017; Risan *et al.*, 2018), and the parameter *A* is typically assumed equal to 105 (Kaimal *et al.*, 1972), later revised to 102 (Kaimal & Finnigan, 1994), and *B* equal to 33."

**Point 12**

*The algorithm in Figure 1 is interesting but also perfectible. It does not clearly show why the iterative procure is necessary and this should be explained in a pedagogical way. For example, it could be stated that there is no need to have an iterative procedure if $f_{th0}$ is equal or lower than 0.01 Hz. There is an argument in favor of the iterative procedure that is not clearly stated in section 2: Choosing a value of $f_{th0}$ too low will result in a poor fit of the velocity spectrum because the number of data points will be reduced. In addition, these points are associated with larger uncertainties than at higher frequencies. At the same time, if $f_{th0}$ is larger than the cut-off frequency, the fitting will be significantly affected by the spatial filtering. There is also a potential limit for the application of this algorithm that was not clearly shown in the manuscript: the spectral correction may fail if the spectral peak is affected by the spatial filtering. Therefore, the proposed method might only be adequate for probe volume of 50 m or lower. That is an issue that deserves further discussion.*

R: We have revised the flowchart of Fig. 1 according to the Reviewer's comments and added more details for the iterative process used for the estimation of $k_{Th}$. At line 137, it is now reported: "First, the pre-multiplied spectrum of the radial velocity projected in the horizontal mean wind direction is fitted with the spectral model of Eq. 1 only for wavenumbers smaller than $k_{Th,0} = 2\pi/l$. Indeed, we expect to observe significant spatial-averaging effects for turbulent length scales smaller than the probe length, $l$. For wavenumbers higher than the selected cut-off value, the ratio between the fitted Kaimal spectrum and the PSD of the LiDAR velocity, $\varphi_*^2$, is calculated to quantify the effect of the energy damping due to the LiDAR measuring process. Subsequently, the LiDAR-to-Kaimal ratio, $\varphi_*^2$, is fitted with Eq. 9 through a least-square algorithm to estimate the filter order, $\alpha$, and provide an updated value for the cutoff wavenumber, $k_{Th}$. This process is iterated until convergence on the parameter $k_{Th}$ is achieved (for this work, the convergence condition imposed is a variation of $k_{Th}$ smaller than 1% of the previous value). If during the iterative process, $k_{Th}$ achieves a value equal or smaller than that corresponding to the spectral peak, $k_p$, then the procedure is arrested and a warning is dispatched indicating that the correction procedure was not successful. This warning condition never occurred for all the data analyzed in this work. Furthermore, it should be considered that when $k_{Th}$ achieves values close to $k_p$, the part of the velocity spectrum, Su, used for the fitting procedure with Eq. 1 can be so limited to jeopardize the accuracy of the fitting procedure.

**Point 13**

*Equation 10: The spatial filtering is a function of the wavenumber rather than the frequency. Therefore, I think that fitting a modified version of Eq. 10, where the frequency is replaced by the wavenumber, may be more appropriate than the original version of Eq. 10.*

R: As recommended by the Reviewer, the filter of Eq. 9, it is now expressed as a function of the wavenumber, $k$.

**Point 14**

*Section 2: is the fitting algorithm a least-square fit?*

R: That is correct. At line 142, it is now reported: "Subsequently, the LiDAR-to-Kaimal ratio, $\varphi_*^2$, is fitted with Eq. 9 through a least-square algorithm to estimate the filter order, $\alpha$…"

**Point 15**
*Line 145: Was the lidar azimuth set manually as equal to the mean wind direction or was it an automated procedure?*
R: For the SLTEST and Celina datasets, the azimuth was set automatically by using the feedback scan modality provided in the software of the Streamline XR LiDAR manufactured by Halo Photonics. At line 174, it is now reported: "… the azimuth angle for the fixed scans was updated automatically at the end of each DBS or VAD scan through the feedback scan mode embedded in the LiDAR software and using the wind-direction value measured at height of 53 m".

**Point 16**
*Line 148: Maybe it can be explained why the sampling frequency was varying between 0.5 Hz and 3.3 Hz?*
R: We thank the Reviewer for this observation. The statement has been revised as follows (line 177): "To investigate possible variations of the averaging process related to the accumulation time, the sampling frequency of the fixed scans was varied between 0.5 Hz and 3.3 Hz, while the range gate was always set equal to 18 m".

**Point 17**
*Figure 2: The topography is a little difficult to see. Maybe you can use a digital terrain model?*
R: We do not aim to provide any specific information about the terrain topography, rather aerial views of the site.

**Point 18**
*Line 163: Since the Obukhov length is calculated, I suggest replacing "static atmospheric stability" by "dynamic atmospheric stability" or simply "atmospheric stability".*
R: For the classification of static and dynamic stability we refer to Sect. 5.5 "Stability Concepts" of the book "An Introduction to Boundary Layer Meteorology" by Ronald B. Stull. Static stability is typically connected to convection and it is governed by the Richardson number, while an example of dynamic instability is the generation of Kelvin-Helmholtz waves, which is a shear-driven instability. For the sake of simplicity, we refer now in the manuscript to atmospheric stability.

**Point 19**
*Line 178: I do not understand the link between the sentence and the reference to Hutchins et al. (2012). Maybe this reference is not necessary?*
R: In Hutchins et al. (2012), the authors used horizontal and vertical arrays of sonic anemometers. To investigate transverse gradients, only data with the 10-minute averaged wind direction within the range ±20° from the direction perpendicular to the horizontal array were considered. The same criterion is now used to reject data with large deviations of the wind direction from the azimuthal angle of the LiDAR.

**Point 20**

*Lines 182-191: These lines could be summarized into a single sentence: "The second-order stationarity is assessed using a moving standard deviation with a window length of 5 min and zero overlapping". The reference to Liu et al is not adequate since they did not invent the concept of moving standard deviation. Besides, I would recommend using overlapping windows for a more robust assessment of the flow stationarity. In Matlab, the function "movstd" can be used for this purpose.*

R: The non-stationary index (IST) is a well-established parameter to investigate the statistical stationarity of time-series, see e.g. Foken *et al.,* 2004. It is not a moving standard deviation, rather a quantification of the percentage variability of the variance over sub-periods of the signal with respect to the variance of the entire signal. It is mathematically different from the moving standard deviation, indeed in Eq. 13, CV is not the signal, rather the variance. In Liu *et al.,* 2017, the IST has been successfully used to select stationary velocity signals collected through sonic anemometers at a site very similar to those involved in this work; hence, it is very relevant for our work.

**Point 21**

*The test of the second-order stationarity is a nice addition by the authors. I would also recommend a test for the first-order stationarity using a moving mean function.*

R: This is an interesting suggestion; however, the standard deviation and turbulence intensity already provide information about the variability of the signal in time over the mean.

**Point 22**

*Line 190: Is there any reason for choosing 40% for the maximal IST value?*

**R:** In Foken et al. (2004) and Liu et al. (2017) a maximum IST of 30% was used. For our work, based on sensitivity analysis, we decided to increase the maximum IST to 40% to enable larger data availability without modifying noticeably the results of our analysis.

**Point 23**

*Line 192: I am not sure I understand the "gradient-based" procedure to remove outliers. Maybe one sentence can be written to make it clearer?*

**R:** More details have been added for the gradient-based filter (line 223): "Specifically, the partial derivative in time of the radial velocity is calculated through a second-order central finite-difference scheme. Velocity samples with absolute partial derivative larger than 15 times the respective median value calculated over the entire signal are marked as outliers and replaced through the inpaint_nans function available in Matlab (D'Errico, 2004). The used threshold value is selected based on a sensitivity analysis".

**Point 24**

*Line 194: There is no official Matlab function "inpaint". However, there exists the function "inpaint_nans" by D'Errico (2004), which is well respected. Is this the function that you were using? If yes, a reference to D'Errico (2004) can be used.*

**R:** The proper name of the function and respective reference are now reported in the manuscript.

**Point 25**

*Eq. 17: This equation is only valid for the mean wind speed. If the variance is computed, substantial errors may arise (see eq. 2 in Sathe and Mann (2012b), which is also valid for a LOS scan mode). Some comments are expected here.*

R: We are thankful to the Reviewer for bringing up this important comment. The along-beam (radial or LOS) velocity variance $\sigma_{V_r}^2$ can be related to the Reynolds stress components as (Eberhard et al., 1989):

$$\sigma_{V_r}^2 = \sigma_u^2 \cos^2 \Phi \cos^2(\theta - \theta_w) + \sigma_v^2 \cos^2 \Phi \sin^2(\theta - \theta_w) + \sigma_w^2 \sin^2 \Phi$$
$$+\sigma_{uv} \sin[2(\theta - \theta_w)] \sin^2 \Phi + \sigma_{uw} \cos(\theta - \theta_w) \sin 2\Phi + \sigma_{vw} \sin 2\Phi \sin(\theta - \theta_w),$$

where $\sigma_u^2, \sigma_v^2, \sigma_w^2$ are the variance of the streamwise, spanwise and vertical velocity components, respectively, and $\sigma_{uv}, \sigma_{uw}$ and $\sigma_{vw}$ are the shear Reynolds stresses. Considering the azimuth angle set in the mean wind direction ($\theta - \theta_w \approx 0$ ensured by constraining the dataset to wind direction variability to $\pm 20°$) and very small elevation angle, the previous equation can be approximated as:

$\sigma_{V_r}^2 \approx \sigma_u^2 + \sigma_v^2(\theta - \theta_w)^2 + \sigma_w^2 \Phi^2 + 2\sigma_{uv}(\theta - \theta_w)\Phi^2 + 2\sigma_{uw}(\theta - \theta_w)\Phi + 2\sigma_{vw}\Phi(\theta - \theta_w).$

Even assuming all the Reynolds stresses with the same magnitude, it is evident that at the first order of approximation $\sigma_{V_r}^2 \approx \sigma_u^2$. Of course, this approximation is not valid for generic values of $\theta$ and $\Phi$. At line 239, it is now added: "Furthermore, the variance of the radial velocity is the first-order approximation of the streamwise-velocity variance given the above-mentioned setup constraints (Eberhard et al., 1989, *Sathe and Mann, 2013*)."

**Point 26**

*Line 228-229: The Savitzky-Golay smoothing filter was not designed by Balasubramaniam (2005) but by Savitzky and Golay (1964). I do not recommend using this filter to smooth the power spectral density (PSD) as it also distorts the low-frequency range of the spectrum. The goal should be to smooth only the high-frequency range, which includes enough data point. I advise to simply bin-average your data over bins that are uniformly spaced in the logarithmic space. This way, the fitting algorithm will also be improved. Alternatively, you can estimate the PSD using auto-regressive methods, which can produce fairly smooth PSD estimates if the random process of interest is broad-banded, which is the case here.*

R: For our data analysis the smoothing with the Savitzky-Golay filter worked as good as bin-averaging, see the figure reported below where the original spectrum is initially high-pass filtered, then smoothed. This procedure does not affect the energy content of the spectral peak. For the smoothing, we used a polynomial function of the second order and the windows are calculated as $int[10(160k)^{0.5}]$, with k reported in 1/m, and *int* is rounding to the closest integer number (Balasubramaniam, 2005). At line 267, it is now reported: "For modeling purpose, the velocity spectra are then smoothed in the wavenumber domain following the Savitztky-Golay filter (Savitzky and Golay , 1964), by using a second-order polynomial function and windows with the width equal to $int[10 (160 \, k)^{0.5}]$, where $k$ is in m$^{-1}$ and *int* is rounding to the closest integer number (Balasubramaniam , 2005)".

[Figure]

**Point 27**

*The method to estimate the velocity spectrum should be explicitly stated. Since a fitting to the velocity spectrum is done, it is important to know if the PSD estimate is reliable or not. I do not recommend using the periodogram method. Alternatives approaches are the modified periodogram method (Welch, 1967) or the multitaper method (Thomson, 1982), both available in Matlab.*

R: For the datasets collected at the Celina and SLTEST sites, the periodogram method has been substituted with the Welch spectrogram. However, very similar results are obtained with both methods. At line 329, it is now reported: "… the PSD of each velocity signal is then calculated with the pwelch function implemented in Matlab (Welch, 1967) without window overlapping and window width corresponding to $k_{co}$".

**Point 28**

*Figure 5: The high-frequency range of the downsampled velocity spectrum seems to be slightly too high. This might be due to the presence of aliasing if the time series were downsampled without filtering. Decimating the original time series with a FIR filter and an order equal to at least 10 is recommended (cf. the Matlab function 'decimate'). This should lead to a better agreement between the velocity spectra of the lidar data and the sonic data.*

R: We are thankful to the Reviewer for this comment and we agree that the down-sampled velocity spectrum might be affected by aliasing. We are now using the function "decimate" to downsample the data collected with the sonic anemometer. At line 285, it is reported: "The horizontal velocity retrieved from the sonic anemometer is first down-sampled with the sampling frequency of the LiDAR measurements, namely 2 Hz, using the Matlab function "decimate" with a finite-impulse response (FIR) low-pass filter with order equal to 10 (Weinstein, 1979)". Figure 5 has been revised accordingly.

**Point 29**

*Line 263-264: The small difference between the difference range gate length is unlikely to explain the difference between the different filter functions. One possible reason for the discrepancies is the presence of measurement noise, which increases with the frequency and which is accounted for in the empirical filter function used in the manuscript but not modelled in eqs. 6-7.*

R: The residual noise present in the LiDAR acquisition surely leads to an overestimation of the high-frequency spectrum. However, we think that it is unlikely that such a big overestimation performed by Eqs. 5 and 6 could be related to the sole measurement noise. At line 306, it is now reported: "A possible explanation for the poor performance of these deconvolution models could be the different probe length used for the XPIA campaign ($l$= 50 m) in contrast to $l$= 30 m used in the original study of that deconvolution model (Mann *et al.*, 2009), and the presence of measurement noise in the data, which is not accounted for in the models of Eqs. 5 and 6".

**Point 30**
*Figure 7 and the associated discussion do not seem to be a vital piece of information here. Firstly, the use of subsamples with an averaging time of 25 s could be criticized as it only includes a small portion of the turbulence spectrum. Therefore, the influence of the correction method on the variance estimate becomes exaggerated, which is not desirable in relation to algorithm validation. Secondly, only one time series is included, which limits the conclusions that can be drawn from this figure. It may be a careful choice to remove this figure and the corresponding paragraph.*
R: This figure and related paragraph have been removed.

**Point 31**
*Line 284-285: When applying a high-pass filter, the reader needs to know the exact value of the cut-off frequency, the order of the filter and what type of filter is applied. Therefore, mentioning a cut-off frequency of the order of $10^{-3}$ Hz is not sufficient. Velocity fluctuations around $1 \times 10^{-3}$ Hz may still be representative of micro-scale turbulence. The use of the high-pass filter will increase the influence of the correction algorithm on the estimation of turbulence characteristics. This can be mentioned and/or quantified.*
R: More details on the used high-pass filter are now reported in the manuscript. The filter cut-off frequency is selected to avoid modifications of the spectral content of the peak. An example of the application of the high-pass filter is reported in the figure of point 26. At line 249, it is now reported: "The LiDAR equivalent velocity, $U_{eq}$, is then high-pass filtered to remove low-frequency non-turbulent velocity fluctuations, using the following spectral transfer function:

$$G(k; S, k_{co}) = \frac{1 + \tanh\left[\beta \log\left(\frac{k}{k_{co}}\right)\right]}{2}$$

where $k_{co}$ is the cutoff wavenumber, which should be smaller than $k_p$ to avoid effects on the spectral peak. The parameter $\beta$ is equal to 100 to generate a sufficiently sharp filter across the cutoff wavenumber, $k_{co}$ (Hu *et al.*, 2019)".

**Point 32**
*Figure 8 includes two subfigures that can be merged into a single one by using a 3-variable scatter plot, where the color of the markers reflects the variance. Nevertheless, I am not sure that figure 8 is vital to the paper.*
R: We initially produced a similar figure to that mentioned by the Reviewer. However, it resulted to be more confusing without saving too much space in the manuscript. Therefore, we would keep this figure to provide a characterization of the background boundary-layer flows.

**Point 33**

*Reference to Guala et al. (2006) is inadequate. They studied turbulence in pipe flows whereas the paper discusses turbulence in the atmospheric boundary layer. A more appropriate reference would be Counihan (1970).*

**R:** That discussion has been removed.

**Point 34**

*Figure 9 can be reduced to a single panel. I think showing the pre-multiplied spectrum $fS_u$ as a function of the wavenumber $k$ is good enough.*

R: The inertial sub-range and the spectral correction is generally more evident through the power spectral density rather than through the pre-multiplied spectra. Plotting the spectra as a function of the reduced frequency, $n$, and wavenumber, $k$, is relevant for the implementation of the procedure. Indeed, more consistent values of the cut-off wavenumber are observed throughout the ASL height, which is not the case for $n$.

**Point 35**

*Line 320: "selected dataset" is not specific enough. Maybe you can mention which data from which campaigns?*

**R:** The sentence has been substituted with (line 369): "The proposed correction of the LiDAR measurements is now applied to all the datasets collected at Celina and SLTEST sites (see Table 2)".

**Point 36**

*Figure 11 looks nice but it is not necessary to the paper. Firstly, because we cannot see a clear difference between the left and right panel and secondly because it does not bring particularly useful information. I suggest removing this figure for the sake of brevity.*

**R:** This figure has been removed.

**Point 37**

*Figure 12 is not necessary either to the paper. The textual description was already enough. I think this figure can be removed.*

**R:** This figure has been removed.

**Point 38**

*Line 330: I am not sure what you mean by a quasi-self-similar behavior. Figure 13 is showing (normalized) transfer functions. I think it may be wise to keep the description as simple as possible.*

R: We mean that the empirical corrections practically collapse on the same curve, which corroborates that the filter proposed in Eq. 9 is actually a good model for the spatial averaging. At line 381, it is now reported: "… all the estimated transfer functions practically collapse on the same curve for measurements collected at different heights".

**Point 39**

*Figure 15 could be replaced by a simple table showing the median value and the interquartile range, for example.*

**R:** The results of Figure 15 are now reported in Table 3 and the discussion at lines 404-414 has been revised accordingly.

**Point 40**

*Section 6 could be reformulated into one or two paragraphs. Firstly, the use of synthetic wind field is not useful here, as calculations are conducted in the frequency domain only. Secondly, the computation of the profiles of the mean wind speed and variance as well as the associated discussion is unnecessary. The right panel of Fig 17 is interesting but does not shows that the error $\epsilon$ could be expressed as a function of the mean wind speed and variance of the velocity only. I suggest shortening in section 6. It is possible to show the dependency of $\varepsilon$ on the mean wind speed and the variance of the velocity as a contour map. Given a reference mean wind speed at a reference height, a logarithmic profile with and a given roughness length and the Kaimal spectrum, constructing such a map is straightforward. To change the mean wind speed parameter, you simply need to change the reference mean wind speed value. To change the variance of the along-wind component, you can simply change the roughness length.*

R: We are greatly thankful to the Reviewer for this highly constructive comment. Sect. 6 has been significantly shortened and the variability of the spatial averaging with friction velocity, aerodynamic roughness length, and sampling height is now reported.

**Point 41**

*The conclusion should be reformulated following the previous comments.*

R: Conclusions have been revised accordingly to the Reviewer's comments.

**Technical corrections**

**Point 1**

*Lines 1-3 (abstract): Maybe you should mention that you are talking about "pulsed" lidar systems. Continuous-wave Doppler wind lidars can measure the flow within a volume much lower than 20 m and a sampling frequency of several hundreds of Hertz.*

R: In the abstract, it is now reported (line 1): "… pulsed wind LiDAR technology…".

**Point 2**

*Line 2 (abstract): a sampling frequency of the order of 10 Hz is mentioned. Do you mean 1 Hz, as written in the manuscript?*

R: In the manuscript, we mention sampling frequency higher than 1 Hz and LiDARs achieving sampling frequency around 10 Hz are now available.

**Point 3**

*Line 3 (abstract): the expression "back-scattered laser beam" may not be correct. Do you mean "backscattered light" or "backscattered signal"?*

R: It is now revised to back-scattered LiDAR signal.

**Point 4**

*Lines 9 (abstract): I suggest replacing "estimated directly from the LiDAR measurement" by a more accurate term: "estimated directly from the power spectral densities of the along-beam velocity component".*

R: This sentence has been revised accordingly.

**Point 5**
*Line 34: The sentence "probe volumes, denoted as range gates" can be misunderstood by the reader and should be reformulated. The range gate length is different from the probe volume. For example, a range gate length shorter than the probe volume implies that the probe volumes are overlapping.*
R: That's correct, the range gate is equal to the probe volume only for non-overlapping probe volumes, which is the case for all the datasets under investigation. The term range gate has been substituted with probe volume or length throughout the manuscript.

**Point 6**
*Line 36: The syntax of the sentence "by means of a laser beam, which is back-scattered" is a little strange. I would write that the light is backscattered but not that the "laser beam" is backscattered.*
R: At line 42, it is now reported: "… utilizing a laser beam, whose light is back-scattered…"

**Point 7**
*Line 38: "from the Doppler shift on the back-scattered signal" should be "from the Doppler shift of the back-scattered signal"*
R: Revised.

**Point 8**
*Line 38: "like those used for the present work" should be "like the one used in the present work".*
R: We used more than one LiDAR and this sentence is grammatically correct.

**Point 9**
*The tilde symbol in Equation 10 and equation 11 may be explicitly defined for the sake of clarity.*
R: At line 129, the following statement has been added: "The symbol $\tilde{}$ is used to differentiate the analytical model of the low-pass filter from its empirical estimate through the ratio between the fitted Kaimal spectrum and the PSD of the LiDAR velocity, $\varphi_*^2$".

**Point 10**
*Line 156: "of the used LiDARs" may be written "of the LiDARs used" instead.*
R: Corrected.

**Point 11**
*Line 175-176: "For the SLTEST and Celina campaigns [...]through DBS or VAD scans" seems to be a repetition of the same information mentioned earlier in the manuscript. Maybe this sentence can be removed.*
R: This sentence has been removed.

**Point 12**
*Line 192: Consider replacing present tense by past tense when describing the data processing.*
R: We typically use the present tense for post-processing and past tense for tasks related to the execution of the experiments and results/tasks from previous works.

**Point 13**
*Line 251: The "spectrum of the lidar signal" should be replaced with "spectrum of the LOS velocity"*
R: Corrected.

**Point 14**
*Line 276: You may replace "linear regression analysis" by "comparison", which is much simpler.*
R: We performed a linear regression, which has a clear mathematical definition to estimate, slope, bias, r-square value, etc.

**Point 15**
*Line 287: The term "first and second-order statistics" can be replaced by "mean value and variance", which are the quantities you study in the paper.*
R: Corrected.

**Point 16**
*Line 307: The sentence "fitting of the LiDAR spectra" should be replaced with "fitting of the Blunt model to the along-beam velocity spectra".*
R: Throughout the manuscript, we use "… fitting with the spectral model of Eq. 1".

**Point 17**
*Line 324: "highest LiDAR gate" should be replaced by "highest LiDAR range gate" or "LiDAR range gate furthest from the instrument".*
R: Revised.

**Point 18**
*Line 336: " [...] always underestimate [...]" Do you mean "overestimate" ?*
R: The analytical models underestimate, it means they should correct more to generate corrected spectra closer to those estimated through the spectral model.

**Point 19**
*Figure 13: The term "convolution function" sounds strange. I would call it "transfer function" as it is the case in the field of signal processing.*
R: Corrected.

**Point 20**
*Line 359: Convolution in the time domain is multiplication in the frequency domain. Therefore, writing that the "synthetic velocity signal is convoluted in the frequency domain" is unclear. I think it may be simpler to write that the velocity spectrum is multiplied with the transfer function modelling the spatial averaging.*
R: This part has been removed.

**References**

Balasubramaniam, B. J. (2005). Nature of turbulence in wall bounded flows. Ph.D. thesis of the University of Illinois at Urbana-Champaign Graduate College.

Banerjee, T., Katul, G. G., Salesky, S. T., & Chamecki, M. (2015). Revisiting the formulations for the longitudinal velocity variance in the unstable atmospheric surface layer. Quarterly Journal of the Royal Meteorological Society, 141(690), 1699–1711.

Bodini, N., Zardi, D., & Lundquist, J. K. (2017). Three-dimensional structure of wind turbine wakes as measured by scanning lidar. Atmospheric Measurement Techniques, 10(8).

Caughey, S. J. (1977). Boundary-layer turbulence spectra in stable conditions. Boundary-Layer Meteorology, 11(1), 3-14.

Cheynet, E., Jakobsen, J. B., Snæbjörnsson, J., Mann, J., Courtney, M., Lea, G., & Svardal, B. (2017). Measurements of surface-layer turbulence in awide norwegian fjord using synchronized long-range doppler wind lidars. Remote Sensing, 9(10), 1–26.

Counihan, J. (1970). Further measurements in a simulated atmospheric boundary layer. Atmospheric Environment (1967), 4(3), 259-275.

D'Errico, J. (2004). Inpaint nans. MATLAB Central File Exchange.

Eberhard, W. L., Cupp, R. E., & Healy, K. R. (1989). Doppler Lidar Measurement of Profiles of Turbulence and Momentum Flux. Journal of Atmospheric and Oceanic Technology, 6, 809-809-819.

Foken, T., Göockede, M., Mauder, M., Mahrt, L., Amiro, B., & Munger, W. (2004). Post-field data quality control. Handbook of micrometeorology, 181-208. Springer, Dordrecht.

Frehlich, R., Hannon, S. M., & Henderson, S. W. (1998). Coherent Doppler lidar measurements of wind field statistics. Boundary-Layer Meteorology, 86(2), 233–256.

Guala, M., Hommema, S. E., & Adrian, R. J. (2006). Large-scale and very-large-scale motions in turbulent pipe flow. Journal of Fluid Mechanics, 554, 521–542.

Held, D. P., & Mann, J. (2018). Comparison of Methods to Derive Radial Wind Speed from a LiDAR Doppler Spectrum. Atmospheric Measurement Techniques Discussion, (August), 1–11.

Højstrup, J. (1981). A simple model for the adjustment of velocity spectra in unstable conditions downstream of an abrupt change in roughness and heat flux. Boundary-Layer Meteorology, 21(3), 341-356.

Højstrup, J. (1982). Velocity spectra in the unstable planetary boundary layer. Journal of the Atmospheric Sciences, 39(10), 2239-2248.

Hu, R., Yang, X. I. A., & Zheng, X. (2019). Wall-attached and wall-detached eddies in wall-bounded turbulent flows. Journal of Fluid Mechanics, 885, A30-24.

Hutchins, N., Chauhan, K., Marusic, I., Monty, J., & Klewicki, J. (2012). Towards reconciling the large-scale structure of turbulent boundary layers in the atmosphere and laboratory. Boundary-Layer Meteorology, 145(2), 273–306.

International Electrotechnical Commission (2007) IEC 61400-1: Wind turbines—part 1: design requirements. 3rd edn.

Iungo, Giacomo Valerio, Yu-Ting Wu, and Fernando Porté-Agel (2013). Field measurements of wind turbine wakes with lidars. Journal of Atmospheric and Oceanic Technology 30(2), 274-287.

Liu, H. Y., Bo, T. L., & Liang, Y. R. (2017). The variation of large-scale structure inclination angles in high Reynolds number atmospheric surface layers. Physics of Fluids, 29(3).

Kaimal, J. C., Wyngaard, J. C., Izumi, Y., & Coté, O. R. (1972). Spectral characteristics of surface-layer turbulence. Quarterly Journal of the Royal Meteorological Society, 98(417), 563–589.

Kaimal, J. C. (1973). Turbulenece spectra, length scales and structure parameters in the stable surface layer. Boundary-Layer Meteorology, 4(1–4), 289–309.

Kaimal, J. C., Wyngaard, J. C., Haugen, D. A., Coté, O. R., Izumi, Y., Caughey, S. J., & Readings, C. J. (1976). Turbulence structure in the convective boundary layer. Journal of the Atmospheric Sciences, 33(11), 2152-2169.

Kaimal, J. C., & Finnigan, J. J. (1994). *Atmospheric boundary layer flows: their structure and measurement*. Oxford university press.

Mann, J., Cariou, J. P., Courtney, M. S., Parmentier, R., Mikkelsen, T., Wagner, R., Enevoldsen, K. (2009). Comparison of 3D turbulence measurements using three staring wind lidars and a sonic anemometer. Meteorologische Zeitschrift, 18(2), 135–140.

Marusic, I., Monty, J., Hultmark, M., & Smits, A. J. (2013). On the logarithmic region in wall turbulence. Journal of Fluid Mechanics, 716(2), R3-1 R3-11.

Monin, A. S., & Obukhov, A. M. (1954). Basic laws of turbulent mixing in the surface layer of the atmosphere. Contrib. Geophys. Inst. Acad. Sci. USSR, 24(151), 163–187.

Panofsky, H. A. (1978). Matching in the convective planetary boundary layer. Journal of the Atmospheric Sciences, 35(2), 272-276.

Risan, A., Lund, J. A., Chang, C. Y., & Sætran, L. (2018). Wind in Complex Terrain-Lidar measurements for evaluation of CFD simulations. Remote Sensing, 10(1), 1–18.

Sathe, A. and Mann, J.: A review of turbulence measurements using ground-based wind lidars, Atmos. Meas. Tech., 6, 3147-3167, https://doi.org/10.5194/amt-6-3147-2013, 2013.

Savitzky, A., & Golay, M. J. (1964). Smoothing and differentiation of data by simplified least squares procedures. Analytical chemistry, 36(8), 1627-1639.

Sjöholm, M., Mikkelsen, T., Mann, J., Enevoldsen, K., & Courtney, M. (2009). Spatial averaging-effects on turbulence measured by a continuous-wave coherent lidar. Meteorologische Zeitschrift, 18(3), 281–287.

Stull, R. B. (1988). An introduction to boundary layer meteorology. Atmospheric sciences.

Taylor, G. I. (1938). The spectrum of turbulence. Proceedings of the Royal Society of London. Series A-Mathematical and Physical Sciences, 164(919), 476-490.

Thomson, D. J. (1982). Spectrum estimation and harmonic analysis. Proceedings of the IEEE, 70(9), 1055-1096

Weinstein, C. J. (1979). Programs for digital signal processing. IEEE.

Welch, P. (1967). The use of fast Fourier transform for the estimation of power spectra: a method based on time averaging over short, modified periodograms. IEEE Transactions on audio and electroacoustics, 15(2), 70-73.

Worsnop, R. P., Bryan, G. H., Lundquist, J. K., & Zhang, J. A. (2017). Using large-eddy simulations to define spectral and coherence characteristics of the hurricane boundary layer for wind-energy applications. *Boundary-Layer Meteorology*, *165*(1), 55-86.

---

## Author Comment (AC2) · 20 Jul 2020

**Spectral correction of turbulent energy damping on wind LiDAR measurements due to  spatial averaging**

Matteo Puccioni and Giacomo Valerio Iungo

Wind Fluids and Experiments (WindFluX) Laboratory, Mechanical Engineering Department, The University of Texas at Dallas, 800 W Campbell Rd, 75080 Richardson, Texas, USA

**Correspondence:** Giacomo Valerio Iungo (valerio.iungo@utdallas.edu)

**Abstract.** Continuous advancements in pulsed wind LiDAR technology have enabled compelling wind turbulence measurements within the atmospheric boundary layer with  probe lengths shorter than 20 m and sampling frequency of the order of 10 Hz. However, estimates of the radial velocity from the back-scattered  LiDAR signal are inevitably affected by an averaging process within each  probe volume, generally modeled as a convolution between the true velocity projected along the LiDAR line-of-sight and an unknown weighting function representing the energy distribution of the laser pulse along the  probe length. As a result, the spectral energy of the turbulent velocity fluctuations is damped within the inertial sub-range, thus not allowing to take advantage of the achieved spatio-temporal resolution of the LiDAR technology. In this article, we propose to correct this turbulent energy damping on the LiDAR measurements by reversing the effect of a low-pass filter, which can be estimated directly from the  power spectral density of the along-beam velocity component. LiDAR data acquired from three different field campaigns are analyzed to describe the proposed technique, investigate the variability of the filter parameters and, for one dataset, assess the procedure for spectral LiDAR correction against sonic anemometer data. It is found that the order of the low-pass filter used for modeling the energy damping on the LiDAR velocity measurements has negligible effects on the correction of the second-order statistics of the wind velocity. In contrast, the cutoff  wavenumber plays a significant role in the spectral correction encompassing the smoothing effects connected with the LiDAR  probe length. Furthermore,  the variability of the spatial averaging on wind LiDAR measurements is investigated for different wind speed, turbulence intensity, and sampling height. The results confirm that the effects of spatial averaging are enhanced with decreasing wind speed, smaller integral length scale and, thus, for smaller sampling height.

**1 Introduction**

Over the last decades, wind Doppler Light Detection and Ranging (LiDAR) technology has provided compelling features to perform wind turbulence measurements within the atmospheric boundary layer (ABL) for different scientific and industrial pursuits, such as air quality, meteorology (Spuler and Mayor , 2005; Emeis et al. , 2007; Bodini et al. , 2017), aeronautic transportation, and wind energy (Frehlich and Kelley , 2008; Zhan et al. , 2019). In the context of ABL turbulence, scanning

25  Doppler wind LiDARs were assessed against other measurement techniques, such as sonic anemometers and scanning Doppler wind radars, during the eXperimental Planetary boundary layer Instrumentation Assessment (XPIA) campaign (Lundquist et al. , 2017; Debnath et al. , 2017a, b; Choukulkar et al. , 2017; Debnath , 2018).

Different scanning strategies can be designed to characterize different properties of the ABL velocity field through LiDAR measurements (Sathe and Mann , 2013), while the highest spectral resolution is achievable by maximizing the sampling fre-

30  quency and measuring over a fixed line-of-sight (LOS). Provided the use of a probe length, $l$, sufficiently small to probe the inertial sublayer at a height from the ground $z$, e.g. $l < 2\pi z$ according to 
[revised manuscript text omitted]

---

## Referee Comment (RC2) · Anonymous Referee #3 · 29 Sep 2020

**General comments**

This manuscript proposes a new method to correct turbulence measurements by Doppler lidar for low-pass filtering due to spatial averaging along the laser path. This is done by a combination of empirical transfer functions with classical Kaimal model spectra. The new approach is similar to commonly-used spectral correction methods for turbulence measurements by sonic anemometers (e.g. Moore, 1986), but at least to my knowledge, this has not be done before in this way for Doppler lidar measurements. The article is generally well written, perhaps a bit lengthy; the structure is clear and the figures are instructive, and the conclusions are drawn correctly. Nevertheless, I have

two main comments, which are somewhat related:

1) Because Kaimal model spectra are based on surface-layer scaling, the proposed correction is only applicable in the surface layer (on the order of 100 m). This important restriction should be made very clear in the manuscript, because modern Doppler lidars often cover the entire boundary layer (on the order of 1000 m), and do not cover the first ca. 50 m. The authors need to clarify this when describing the scope of this study in the introduction section and also in the discussion and conclusion sections.

2) The literature review in the introduction section is incomplete. Particularly, one important reference is missing (Brugger et al., 2016), which has the same objective of proposing a spectral correction method for Doppler lidars to compensate for spatial averaging along the laser path. However, the underlying approach is very different (Frehlich and Cornman, 2002), which is based on a von Kármán turbulence model. This has the advantage that it is independent of stratification or the driving mechanism of turbulence. The authors need to explain what the differences between the two approaches are, theoretical and practical, disadvantages and advantages, and they also need to justify why they propose this new method. This affects the introduction and also the discussion and conclusion sections.

References

Brugger, P., Träumner, K. and Jung, C.: Evaluation of a procedure to correct spatial averaging in turbulence statistics from a doppler lidar by comparing time series with an ultrasonic anemometer, J. Atmos. Ocean. Technol., 33(10), 2135–2144, doi:10.1175/JTECH-D-15-0136.1, 2016.

Frehlich, R. and Cornman, L.: Estimating spatial velocity statistics with coherent Doppler lidar, J. Atmos. Ocean. Technol., 19(3), 355–366, doi:10.1175/1520-0426-19.3.355, 2002. Moore, C. J.: Frequency response corrections for eddy correlation systems, Boundary-Layer Meteorol., 37(1–2), 17–35, doi:10.1007/BF00122754, 1986.

Moore, C. J.: Frequency response corrections for eddy correlation systems, Boundary-Layer Meteorol., 37(1–2), 17–35, doi:10.1007/BF00122754, 1986.

---

## Author Comment (AC3) · 27 Oct 2020

**Reply to the comments provided by Anonymous Referee #3 on the manuscript amt-2020-27 entitled "Spectral Correction of turbulent energy damping on wind LiDAR measurements due to range-gate averaging", by M. Puccioni and G. V. Iungo**

The authors are greatly thankful to the Reviewer for the thorough review and insightful comments. Our replies are reported in the following. References to pages and lines correspond to those of the latest marked-up manuscript.

**General comments**

*This manuscript proposes a new method to correct turbulence measurements by Doppler lidar for low-pass filtering due to spatial averaging along the laser path. This is done by a combination of empirical transfer functions with classical Kaimal model spectra. The new approach is similar to commonly-used spectral correction methods for turbulence measurements by sonic anemometers (e.g. Moore, 1986), but at least to my knowledge, this has not be done before in this way for Doppler lidar measurements. The article is generally well written, perhaps a bit lengthy; the structure is clear and the figures are instructive, and the conclusions are drawn correctly. Nevertheless, I have two main comments, which are somewhat related:*

R: We thank the Reviewer for the positive comments on our research strategy and the results achieved.

**Comments**

*1)       Because Kaimal model spectra are based on surface-layer scaling, the proposed correction is only applicable in the surface layer (on the order of 100 m). This important restriction should be made very clear in the manuscript, because modern Doppler lidars often cover the entire boundary layer (on the order of 1000 m), and do not cover the first ca. 50 m. The authors need to clarify this when describing the scope of this study in the introduction section and also in the discussion and conclusion sections.*

**R:** The authors are thankful to the Referee for this observation. We have highlighted throughout the manuscript that the proposed method only applies to LiDAR data collected within the surface layer. In the Abstract, L21: "On the other hand, the proposed method assumes that surface-layer similarity holds". In the Introduction, L 87, it is now reported: "It is noteworthy that this method leverages surface layer similarity (Stull, 1988), thus it can only be applied for wind LiDAR measurements collected within the ASL". At L 182: "On the other hand, the proposed procedure leverages the surface-layer similarity for the Kaimal spectral model for the streamwise model and, thus, it can only be applied for wind LiDAR measurements collected within the ASL". In the Conclusions, L 556: "It is noteworthy that the Kaimal spectral model leverages surface layer similarity and, thus, the proposed method can only be used for LiDAR measurements collected within the atmospheric surface layer (ASL)".

*2)       The literature review in the introduction section is incomplete. Particularly, one important reference is missing (Brugger et al., 2016), which has the same objective of proposing a spectral correction method for Doppler lidars to compensate for spatial averaging along the laser path. However, the underlying approach is very different (Frehlich and Cornman, 2002), which is based on a von Kármán turbulence model. This has the advantage that it is independent of stratification or the driving mechanism of turbulence. The authors need to explain what the differences between*

*the two approaches are, theoretical and practical, disadvantages and advantages, and they also need to justify why they propose this new method. This affects the introduction and also the discussion and conclusion sections.*

**R:** The authors are thankful to the Referee for this comment. The paper Brugger et al. (2016) is now reviewed in the Introduction. Furthermore, this method is compared to the proposed method and other two existing methods for one LiDAR dataset in the new Fig. 12. The advantages/disadvantages of the new method compared to existing methods are now better highlighted throughout the manuscript. In the Abstract, L19, it is now reported: "
[revised manuscript text omitted]

---

## Referee Report (RR1)

**Spectral correction of turbulent energy damping on wind LiDAR measurements due to range-gate averaging**

Reviewer's comments

**1 General comment**

The revised manuscript "Spectral correction of turbulent energy damping on wind LiDAR measurements due to range-gate averaging" by Puccioni and Iungo has been significantly improved. The fact they take the time to answer my numerous comments is praiseworthy and highlight their dedication to improving the manuscript. Nevertheless, there remain some few crucial aspects that need to be addressed before the manuscript can be accepted for publication in Atmospheric Measurement Techniques. I list them in the section hereafter.

**2 Specific comments**

**Point 1**

The authors mentioned that the revised manuscript is significantly shortened but I see that the second version is longer than the first one. I invite the authors to assess which elements are superficial to make the manuscript more compact, if possible.

**Point 2**

In the reply to my comment on point 20, the authors state that their stationary test is not a "moving standard deviation" but a "quantification of the percentage variability of the variance over sub-periods of the signal with respect to the variance of the entire signal". I would like to apologize if my original comment was unclear because the second-order stationary test I mentioned in my original point is what the authors described in their reply. More precisely, their test is the application of an archaic moving standard deviation function normalized by the standard deviation of the signal. I recommend the author not to use jargon term (IST) and refer to an older source than Liu et al (2017), for example, Foken and Wichura (1996). Therefore, my original suggestion "The second-order stationarity is assessed based on a moving standard deviation function with a window length of 5 min and zero overlappings" is still adequate. The authors can also complement this description by stating that they assessed the relative error $\varepsilon$ of the moving standard deviation with respect to the standard deviation of the entire signal. This relative error is what they call IST, except it is a longer and more ambiguous term. Note that the test by Foken and Wichura (1996) is also outdated because there is no overlapping and relies on numerical methods developed in the 1980s during

which the computational power was limited. I trust the authors regarding the reliability of their test. Nevertheless, it is important to note that contrary to what Foken and Wichura (1996) state, this test should not be used to assess the stationarity of the friction velocity if the window length is only 5 min long with a threshold value $\varepsilon_T$ of 30% (or even 40%) for the relative difference. The reason is that the random error associated with the estimation of the friction velocity using an averaging time of only 5 min is likely above 50% (Kaimal and Finnigan, 1994). In this situation, the stationary the test will fail to distinguish random error from systematic bias. For the variance of the signal, the random error is much lower so the value $\varepsilon_T = 40\%$ chosen by the authors is likely appropriate.

**Point 3**

The point 21 in my previous review aimed to highlight that assessing the second-order stationarity without looking at the first-order stationarity makes little sense for spectral analysis. Fortunately for the manuscript, I am not expecting significant changes in the results since the second-order stationarity is more difficult to achieve than the first one. Nevertheless, this step cannot be neglected and has to be addressed. Contrary to what the authors claim, the turbulence intensity and standard deviation cannot inform about the signal stationarity since they are only applicable to stationary random processes.

**Point 4**

In your reply to point 27, you wrote that the periodogram method was replaced with Welch's algorithm and that similar results were obtained. In Matlab, using Welch's algorithm without overlapping and a single window is similar to applying the periodogram method but with a Hamming window instead of a rectangular window. It seems that in your data analysis, you have applied a single window (I might be wrong here). Therefore, this might explain the lack of substantial improvement in the power spectral density (PSD) estimates. Nevertheless, the periodogram method is known to be a poor power spectral density estimator. I recommend trying two to three segments with Welch's algorithm and 50% overlapping. This could significantly change the outcome of your fitting algorithm.

**Point 5**

In your reply to point 29, the authors wrote that measurement noise in the lidar velocity records is unlikely to produce the big overestimation observed in Fig 5. I understand that my initial suggestion was maybe too vague. Therefore, I have reproduced such an overestimation in fig. 1 by filtering an idealized velocity spectrum using functions which emulates the presence or absence of noise at high frequencies. I have actually applied two low-pass filters with a different order for the sake of simplicity. In the left panel of fig. 1, one could assume that the red curve has a larger amplitude than the blue one because of measurement noise. The correction of the velocity spectrum is done by using the ratio $H(f_r)$ between the black curve and the blue curve. Therefore, the blue curve is properly corrected. However, the noise in the red curve is massively amplified when applying the same correction, because $H(f_r)$ erroneously assumed no measurement noise.

[Figure]

Figure 1: Left: Velocity spectra affected by spatial averaging with and without the presence of noise. Right: Amplification of the noise level by correction the velocity spectra using an idealized spectral transfer function.

The right panel of fig. 1 shows similar behaviour as in the right panel of your Fig. 5. Even if the random noise is small, the application of an idealized spectral correction will likely produce a velocity spectrum with an unacceptable level of noise in the high-frequency range. The misalignment of the scanning beam with the instantaneous wind direction might also contribute to the overestimation of the corrected power spectral density in the high-frequency range. As stated by the authors, the different probe volume length could also affect the high frequencies velocity fluctuations, but I am not sure to what extent. I believe that different probe volume lengths are likely to affect more the low-frequency velocity fluctuations than the high-frequency ones. In this regards, your correction algorithm outperforms those based on an idealized spatial filtering function because it accounts for the random error in the spatially filtered velocity spectrum.

**References**

Foken, T., Wichura, B.. Tools for quality assessment of surface-based flux measurements. Agricultural and forest meteorology 1996;78(1-2):83–105.

Kaimal, J.C., Finnigan, J.J.. Atmospheric boundary layer flows: their structure and measurement. Oxford university press; 1994.

---

## Author Response (AR2)

**Reply to the comments provided by the Reviewer on the manuscript amt-2020-27 entitled "Spectral Correction of turbulent energy damping on wind LiDAR measurements due to range-gate averaging", by M. Puccioni and G. V. Iungo**

The authors are greatly thankful to the Reviewer for the thorough review and the insightful comments. Our replies are reported in the following. References to pages and lines correspond to those of the latest marked-up manuscript.

**1. General comment**

*The revised manuscript "Spectral correction of turbulent energy damping on wind LiDAR measurements due to range-gate averaging" by Puccioni and Iungo has been significantly improved. The fact they take the time to answer my numerous comments is praiseworthy and highlight their dedication to improving the manuscript. Nevertheless, there remain some few crucial aspects that need to be addressed before the manuscript can be accepted for publication in Atmospheric Measurement Techniques. I list them in the section hereafter.*

**R**: We are thankful to the Reviewer for his/her positive feedback and for the insightful review. The provided recommendations have been extremely valuable to improve our work.

**2. Specific comments**

**Point 1**

*The authors mentioned that the revised manuscript is significantly shortened but I see that the second version is longer than the first one. I invite the authors to assess which elements are superficial to make the manuscript more compact, if possible.*

**R**: After the first review, we shortened the manuscript where possible. However, a new analysis has been added at Sect. 6 (Fig. 13) thanks to the suggestions provided by the Reviewer.

**Point 2**

*In the reply to my comment on point 20, the authors state that their stationary test is not a "moving standard deviation" but a "quantification of the percentage variability of the variance over sub-periods of the signal with respect to the variance of the entire signal". I would like to apologize if my original comment was unclear because the second-order stationary test I mentioned in my original point is what the authors described in their reply. More precisely, their test is the application of an archaic moving standard deviation function normalized by the standard deviation of the signal. I recommend the author not to use jargon term (IST) and refer to an older source than Liu et al (2017), for example, Foken and Wichura (1996). Therefore, my original suggestion "The second-order stationarity is assessed based on a moving standard deviation function with a window length of 5 min and zero overlapping" is still adequate. The authors can also complement this description by stating that they assessed the relative error ε of the moving standard deviation with respect to the standard deviation of the entire signal. This relative error is what they call IST, except it is a longer and more ambiguous term. Note that the test by Foken and Wichura (1996) is also outdated because there is no overlapping and relies on numerical methods developed in the 1980s during which the computational power was limited. I trust the authors regarding the reliability of their test. Nevertheless, it is important to note that contrary to*

*what Foken and Wichura (1996) state, this test should not be used to assess the stationarity of the friction velocity if the window length is only 5 min long with a threshold value $\varepsilon_T$ of 30% (or even 40%) for the relative difference. The reason is that the random error associated with the estimation of the friction velocity using an averaging time of only 5 min is likely above 50% (Kaimal and Finnigan, 1994). In this situation, the stationary the test will fail to distinguish random error from systematic bias. For the variance of the signal, the random error is much lower so the value $\varepsilon_T = 40\%$ chosen by the authors is likely appropriate.*

**R**: We are thankful to the Reviewer for these further comments. The reference to Liu et al. (2017) has been removed and replaced by Foken and Wichura (1996). The quantity previously indicated with IST is now referred to as $\epsilon_\sigma$. In the text at L 219, it is now reported: "A similar parameter $\varepsilon_\sigma$ is calculated for the second-order statistics by (Foken and Wichura, 1996):

$$\epsilon_\sigma = \frac{\frac{1}{N}\sum_{j=1}^{N} CV_j - CV_{tot}}{U_{tot}},$$

where $CV_j$ is the variance of the j-th sub-period with a duration of 5 minutes, while $CV_{tot}$ is the variance of the signal over the entire period…".

**Point 3**

*The point 21 in my previous review aimed to highlight that assessing the second-order stationarity without looking at the first-order stationarity makes little sense for spectral analysis. Fortunately for the manuscript, I am not expecting significant changes in the results since the second-order stationarity is more difficult to achieve than the first one. Nevertheless, this step cannot be neglected and has to be addressed. Contrary to what the authors claim, the turbulence intensity and standard deviation cannot inform about the signal stationarity since they are only applicable to stationary random processes.*

**R**: As mentioned by the Reviewer, this check is trivial being the stationarity of the mean a weaker constraint than the stationarity of the second-order statistics. In analogy to the relative error of the variance, a relative absolute error for the mean is calculated as follows and reported in Table 2:

$$\varepsilon_M = \frac{\frac{1}{N}\sum_{j=1}^{N}\left|U_j - U_{tot}\right|}{U_{tot}},$$

where $U_j$ is the mean velocity calculated over a 5-minute sub-period, $N$ is the number of non-overlapping sub-periods, and $U_{tot}$ is the mean of the entire velocity signal. For each dataset, the maximum relative absolute error for the mean is reported in the following table and in Table 2:

| Name | $\epsilon_M [\%]$ |
|---|---|
| XPIA | 9.56 |
| SLTEST | 6.57 |
| Celina1 | 4.81 |
| Celina2 | 11.50 |
| Celina3 | 11.67 |

At L 214, it is now reported: "The statistical steadiness of the LiDAR signals is estimated for both first- and second-order statistics. For the mean velocity, the absolute percentage error is calculated as follows:

$$\varepsilon_M = \frac{\frac{1}{N}\sum_{j=1}^{N}\left|U_j - U_{tot}\right|}{U_{tot}}$$

where $U_j$ is the mean velocity as a function of height, z, calculated for the j-th sub-period with a duration of 5 minutes, sub-period, $U_{tot}$ is the mean wind velocity as a function of height for the entire velocity signals, while $N$ is the total number of sub-periods generated without overlapping". At L 224, it is reported: "For quality control purpose, signals with $\epsilon_M \geq 15\%$ or $\epsilon_\sigma \geq 40\%$ are usually rejected (Foken and Wichura, 1996; Foken et al., 2004). The parameters $\epsilon_M$ and $\epsilon_\sigma$ are calculated for each range gate and their maximum values for the selected datasets are reported in Table 2".

**Point 4**: *In your reply to point 27, you wrote that the periodogram method was replaced with Welch's algorithm and that similar results were obtained. In Matlab, using Welch's algorithm without overlapping and a single window is similar to applying the periodogram method but with a Hamming window instead of a rectangular window. It seems that in your data analysis, you have applied a single window (I might be wrong here). Therefore, this might explain the lack of substantial improvement in the power spectral density (PSD) estimates. Nevertheless, the periodogram method is known to be a poor power spectral density estimator. I recommend trying two to three segments with Welch's algorithm and 50% overlapping. This could significantly change the outcome of your fitting algorithm.*

**R**: We are thankful to the Reviewer for this further comment. As reported in the manuscript, before the calculation of the power spectral density, each signal is high-pass filtered with a cut-off wavenumber $k_{co} = 10^{-3}\text{m}^{-1}$ (L 317). Subsequently, the power spectral density for the Celina and SLTEST data are calculated with the Welch's algorithm without window overlapping and with a sub-window length inversely proportional to $k_{co}$. In particular, the number of sampling points within a single window is given by:

$$n_w = \text{floor}\left(\frac{f_S}{f_{co}}\right),$$

where $f_S$ is the sampling rate and $f_{co}$ is the detrending frequency calculated from $k_{co}$ through Taylor's frozen-turbulence hypothesis (Taylor, 1938). Therefore, the number of non-overlapping windows has been calculated as:

$$N_w = \text{floor}\left(\frac{N_{TOT}}{n_w}\right),$$

were $N_{TOT}$ is the total number of samples of the velocity signal. The number of windows for each dataset is reported in the following table:

| Name | $N_w$ |
| --- | --- |
| XPIA | 1 |
| SLTEST | 23 |
| Celina1 | 8 |
| Celina2 | 9 |
| Celina3 | 9 |

For the XPIA dataset, only one window has been used due to the short duration of the velocity acquisition (14.5 minutes) and the need to characterize the lowest frequency available. Finally, a smoothing procedure in the wavenumber domain is applied following the Savitztky-Golay filter (L270). As pointed out by the Reviewer, the specific algorithm used to evaluate the spectra may affect the result of the correction procedure. In the manuscript, it is stated that the main parameter determining the amount of variance recovery is the cutoff wavenumber, $k_{Th}$. Thus, we conducted

a sensitivity analysis of $k_{Th}$ on the algorithm used for the calculation of the spectra. For this analysis, we leveraged the SLTEST dataset. In particular, for each LiDAR gate, the spectrum of the velocity signal is calculated through the following algorithms:

1. Welch algorithm with 3 windows and 50% overlapping;
2. Welch algorithm with 3 windows, 50% overlapping and smoothing;
3. Welch algorithm with $N_w$ windows and no overlapping;
4. Welch algorithm with $N_w$ windows, no overlapping and smoothing (as done in the manuscript).

An example of the resulting spectra is reported in the figure below for the LiDAR signal collected at z= 10 m.

[Figure]

Subsequently, the correction procedure is carried out for each LiDAR velocity signal and spectral procedure. It is evident that the resulting cutoff wavenumber is practically insensitive to the different spectral procedures used.

[Figure]

**Point 5**

*In your reply to point 29, the authors wrote that measurement noise in the lidar velocity records is unlikely to produce the big overestimation observed in Fig 5. I understand that my initial suggestion was maybe too vague. Therefore, I have reproduced such an overestimation in fig. 1 by filtering an idealized velocity spectrum using functions which emulates the presence or absence of noise at high frequencies. I have actually applied two low-pass filters with a different order for the sake of simplicity. In the left panel of fig. 1, one could assume that the red curve has a larger amplitude than the blue one because of measurement noise. The correction of the velocity spectrum is done by using the ratio H(f_r) between the black curve and the blue curve. Therefore, the blue curve is properly corrected. However, the noise in the red curve is massively amplified when applying the same correction, because H(f_r) erroneously assumed no measurement noise.*

*The right panel of fig. 1 shows similar behavior as in the right panel of your Fig. 5. Even if the random noise is small, the application of an idealized spectral correction will likely produce a velocity spectrum with an unacceptable level of noise in the high-frequency range. The mis-alignment of the scanning beam with the instantaneous wind direction might also contribute to the overestimation of the corrected power spectral density in the high-frequency range. As stated by the authors, the different probe volume length could also affect the high frequencies velocity fluctuations, but I am not sure to what extent. I believe that different probe volume lengths are likely to affect more the low-frequency velocity fluctuations than the high-frequency ones. In this regards, your correction algorithm outperforms those based on an idealized spatial filtering function because it accounts for the random error in the spatially filtered velocity spectrum.*

**R**: We agree with the Reviewer about the impact of the instrument noise on the high-frequency portion of the spectrum. In the manuscript at L 304, it is now reported: "A possible explanation for the poor performance of these deconvolution models could be the presence of residual noise in the data, which is not accounted for in the models of Eq. 5 and 6. Other factors, like the different probe length used for the XPIA campaign ($l = 50$m) in contrast to $l = 30m$ used in the original study of this deconvolution model (Mann et al., 2009), are thought to have marginal effects on the estimation of the energy spectrum at higher frequencies."

**References**:**

Taylor, G. I. (1938). The spectrum of turbulence. *Proceedings of the Royal Society of London. Series A-Mathematical and Physical Sciences*, *164*(919), 476-490.

[revised manuscript text omitted]